# Simultaneous magnetic resonance imaging of pH, perfusion and renal filtration using hyperpolarized $^{13}$C-labelled Z-OMPD

Martin Grashei [1,8], Pascal Wodtke [1,8], Jason G. Skinner[1], Sandra Sühnel[1], Nadine Setzer[1], Thomas Metzler [2], Sebastian Gulde [3], Mihyun Park [4], Daniela Witt[4], Hermine Mohr [3], Christian Hundshammer[1], Nicole Strittmatter [4], Natalia S. Pellegata[3,5], Katja Steiger [2] & Franz Schilling [1,6,7] ✉

pH alterations are a hallmark of many pathologies including cancer and kidney disease. Here, we introduce [1,5-$^{13}$C$_2$]Z-OMPD as a hyperpolarized extracellular pH and perfusion sensor for MRI which allows to generate a multiparametric fingerprint of renal disease status and to detect local tumor acidification. Exceptional long $T_1$ of two minutes at 1 T, high pH sensitivity of up to 1.9 ppm per pH unit and suitability of using the $C_1$-label as internal frequency reference enables pH imaging in vivo of three pH compartments in healthy rat kidneys. Spectrally selective targeting of both $^{13}$C-resonances enables simultaneous imaging of perfusion and filtration in 3D and pH in 2D within one minute to quantify renal blood flow, glomerular filtration rates and renal pH in healthy and hydronephrotic kidneys with superior sensitivity compared to clinical routine methods. Imaging multiple biomarkers within a single session renders [1,5-$^{13}$C$_2$]Z-OMPD a promising new hyperpolarized agent for oncology and nephrology.

Onco-nephrology is an emerging medical field in the interdisciplinary care of patients to identify and prevent kidney damage or failure related to cancer treatments as well as to assess risk for cancer due to kidney disease[1,2]. Upon diagnosis of cancer, state-of-the-art treatment options employ chemo- and radiotherapy. While treatment outcome can be significantly affected by prevailing pH conditions in the tumor[3], pH-regulating adjuvant therapies[4] might be crucial to enhance patient outcome[5], as long and ineffective therapies often pose a severe kidney burden[6]. Therefore, for many therapies it might be essential to assess the tumor pH, as well as the renal function before, during and following cancer treatment, for which renal acid-base balance and glomerular filtration rates[7] are important biomarkers. In clinical routine, kidney function is probed by iodinated contrast multiphase CT[8,9] or scintigraphy using $^{99m}$Tc-MAG3[10] as imaging agent. Recently introduced methods, not translated in clinical routine yet, involve the injection of Gd-based contrast agents for DCE-MRI[11], diffusion weighting[12] or arterial spin labelling[13,14]. However, scintigraphy using $^{99m}$Tc-MAG3 involves the injection of ionizing radiation and spin-labelling- and diffusion-weighted-techniques are sensitive to abdominal motion and limit assessment of the complete renal filtration

[1]Department of Nuclear Medicine, TUM School of Medicine, Klinikum rechts der Isar, Technical University of Munich, D-81675 Munich, Germany. [2]Comparative Experimental Pathology (CEP), Institute of Pathology, School of Medicine, Technical University of Munich, D-81675 Munich, Germany. [3]Institute for Diabetes and Cancer, Helmholtz Zentrum München, D-85764 Neuherberg, Germany. [4]Department of Biosciences, TUM School of Natural Sciences, Technical University of Munich, D-85748 Garching, Germany. [5]Department of Biology and Biotechnology, University of Pavia, I-27100 Pavia, Italy. [6]Munich Institute of Biomedical Engineering, Technical University of Munich, D-85748 Garching, Germany. [7]German Cancer Consortium (DKTK), Partner Site Munich and German Cancer Research Center (DKFZ), D-69120 Heidelberg, Germany. [8]These authors contributed equally: Martin Grashei, Pascal Wodtke. ✉e-mail: schilling@tum.de

process. Further, CT-based contrast agents can induce nephropathies[15], which poses a major risk for single or repeated assessment of already fragile kidney conditions, while MRI-based contrast agents can accumulate in organs[16], bear a risk of inducing nephrogenic systemic fibrosis in patients with impaired kidney function[17] and are suspected to accelerate metastasis[18]. In addition, despite several first studies to image pH in human patients[19–23], there is no routinely applied noninvasive imaging method available yet. These approaches rely on iodine-based injectable CEST-agents for extra-cellular pH measurements (acidoCEST)[19,21–23] or generate only pH- and intracellularly-weighted images using endogenous amide protons for contrast generation (APT-CEST)[20]. Simultaneously, there is a strong clinical need for safe and fast imaging of extracellular pH to assess tumor acidification for patient stratification, efficacy of adjuvant therapies or early response to therapy[24].

Hyperpolarized magnetic resonance imaging is an imaging technique on the transition to the clinic[25], which relies on the injection of isotopically enriched, non-toxic contrast agents. Among these, [$^{13}$C]urea[26], [$^{13}$C, $^{15}$N$_2$]urea[27], [$^{13}$C]2-Methylpropan-2-ol[28], HP001[29] and hyperpolarized water[30] have been introduced as perfusion agents. While the latter three have been preclinically investigated for assessment of kidney[27,31–33] and tumor[34] perfusion with quantitative assessment of kidney filtration rates[35], [$^{13}$C, $^{15}$N$_2$]urea has also been recently translated into clinical trials[36,37]. Further, only [1-$^{13}$C]pyruvate has so far been applied in an onco-nephrological setting to image renal cell carcinoma[38,39]. For non-invasive assessment of pH, both in renal function and cancer, hyperpolarized [$^{13}$C]bicarbonate[40] and [1,5-$^{13}$C$_2$] zymonic acid[41] have been introduced preclinically but have not been clinically translated yet, only the latter providing exclusively extra-cellular pH information. A comprehensive assessment of kidney function requires simultaneous imaging of at least perfusion and pH. Current efforts for combined imaging protocols are, however, limited either by a time-consuming combination of different imaging modalities[42] or potentially nephrotoxic contrast agents[7] and so far haven't been clinically translated.

In this work, we introduce hyperpolarized, $^{13}$C-labelled [1,5-$^{13}$C$_2$]Z-4-methyl-2-oxopent-3-enedioic acid (Z-OMPD) as a probe for simultaneous 3D imaging of renal perfusion and filtration and 2D imaging of pH in vivo within a single injection. In addition, the C$_1$-resonance allows internal chemical shift referencing for pH determination with a so far unmet spatial resolution for hyperpolarized pH imaging in kidneys and tumors. This molecule, which has been shown to naturally occur in tulips[43] and is assumed to occur in interstellar meteorites[44], allows to read-out renal blood flow, glomerular filtration rates and kidney pH compartments by selectively utilizing the hyperpolarization of its $^{13}$C$_1$- and $^{13}$C$_5$-resonances. The combined, multiparametric imaging technique was then applied to a clinically relevant hydronephrosis model[45] induced by oversecretion of catecholamines from pheochromocytoma, where it demonstrates superior sensitivity in detecting pathological kidney alterations compared to safe clinical routine methods.

## Results

### Synthesis of Z-OMPD
Given the lack of non-degrading, easy-to-polarize, chemical shift-based and long-lived hyperpolarized $^{13}$C-labelled pH sensors[46], recently emerging synthesis routes and structures of pyruvic acid-derived molecules[47] were analyzed for molecules with favourable hyperpolarization properties and T$_1$ relaxation times[48]. Here, out of this molecule class, previously introduced [1,5-$^{13}$C$_2$]zymonic acid[41] exhibits a pK$_a$ in the physiological range and good hyperpolarization properties, but requires co-polarization and degrades. We identified Z-OMPD as the pyruvic acid dimer with the most favourable pH-sensing properties having the lowest energy level of OMPD isomers and high biocompatibility[43,47,49]. The C$_1$- and C$_5$-positions were considered to exhibit the most promising pH sensitivity and relaxation properties.

The recently published synthesis route[47] was modified by using [1-$^{13}$C]ethyl pyruvate as a precursor to synthesize [1,5-$^{13}$C$_2$]Z-OMPD (Fig. 1a, for details see methods). Product purity was improved by the development of a HPLC-based purification protocol with a final yield of ~18% (weight fraction) and a purity of ≥96% (Fig. S1). The structure and mass of unlabelled Z-OMPD and $^{13}$C-labelled [1,5 $^{13}$C$_2$]Z-OMPD were confirmed by $^1$H-NMR spectroscopy and mass spectrometry (Fig. S2), respectively.

### pH sensitivity
To evaluate the pH sensitivity of the $^{13}$C-labelled carboxylic acid carbons, a titration series of thermal $^{13}$C-NMR spectra in the in vivo relevant pH range (~pH 7) reveals changes of the chemical shift of both $^{13}$C-labels with pH in opposite field directions (Fig. 1b) relative to [13C]urea as a chemical shift reference. Notably, the $^{13}$C$_5$-resonance exhibits a significantly stronger pH dependence (3.0 ppm over the whole pH range) compared to the $^{13}$C$_1$-resonance (0.4 ppm), which shows only a weak pH sensitivity. A calibration curve for the peak positions relative to [$^{13}$C]urea for chemical shift-based pH measurements was obtained by fitting a scaled logistic function (Fig. 1c). The resulting acid dissociation constant pK$_{a2}$ = 6.55, together with the dynamic range of the pH-sensitive chemical shift of C$_5$ renders [1,5-$^{13}$C$_2$] Z-OMPD a suitable pH sensor for the physiological and pathological pH range around pH 7.4. Furthermore, Z-OMPD's pH sensing accuracy is not compromised by variations in probe- and protein concentration, ionic strength and temperature, which altogether lead to uncertainties of less than 0.05 pH units in a physiological setting (Fig. S3).

### Chemical and thermal stability
Hyperpolarized $^{13}$C-MRI of Z-OMPD requires chemical and thermal stability, which affect sample storage, preparation, hyperpolarization protocols, as well as safety and functionality in vivo. Probe stability in D$_2$O and during heating was tested. Z-OMPD proved to be stable in aqueous solutions for at least 15 days (Fig. S4a) with no indication for any decay products. Neither heating to 62 °C for one hour nor additional sonication for 1 hour showed any signs of degradation of Z-OMPD (Fig. S4b).

### Hyperpolarization properties
The transient nature of the hyperpolarized state requires probes to exhibit a high polarization level and a long T$_1$ relaxation time constant. For Z-OMPD, polarization levels between 17% and 26% were reached after 1.5 h of polarization for both $^{13}$C-labels (see Methods). T$_1$ relaxation time constants for both $^{13}$C-labels turned out to be significantly longer (C$_1$: 138 s, C$_5$: 119 s) than those of alternative hyperpolarized in vivo pH sensors like [$^{13}$C]bicarbonate[40] or [1,5-$^{13}$C$_2$,3,6,6,6-D$_4$]zymonic acid (ZA$_d$; C$_1$: 47 s, C$_5$: 75 s) (Fig. 1d). Measurements in D$_2$O at 1 T (n = 2) and 7 T (n = 1), blood at 1 T (n = 1) and 7 T (n = 2), as well as in vivo (n = 2) showed long T$_1$ relaxation times up to two minutes for both $^{13}$C-labels in vitro and half a minute in vivo at 7 T (Fig. 1e), where the latter also strongly exceeds those of zymonic acid (C$_1$: 17 s, C$_5$: 16 s[41]) or bicarbonate (T$_1$ = 10.1 s[40]).

### pH imaging in phantoms
pH imaging was first tested in two different in vitro phantom setups. pH maps from chemical shift imaging (CSI) acquisitions of six hexagonally oriented citrate-phosphate buffer phantoms of different pH shows clear pH contrast between and good uniformity within each phantom while agreeing with electrode measurements (Fig. 2a). Imaging of three blood samples of different pH also shows good pH precision and accuracy (Fig. 2b). In addition, pH values measured using the electrode pH and the spectroscopically derived pH using Z-OMPD were compared at different buffer compositions, in blood as well as for varying Z-OMPD-, salt- and protein concentrations (Fig. 2c). Linear regression shows very good agreement between both techniques across the entire probed pH range (r = 0.99).

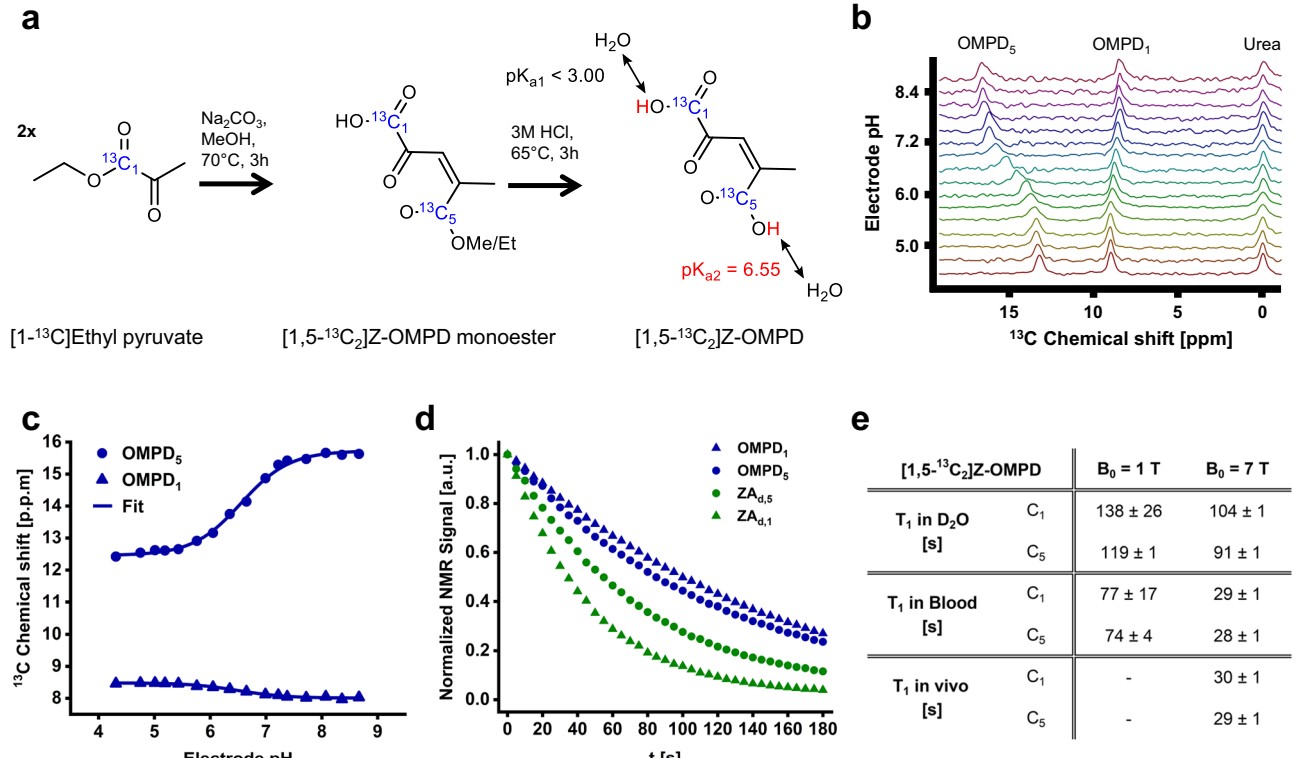

**Fig. 1 | Synthesis and characterization of [1,5-$^{13}$C$_2$]Z-OMPD. a** Z-OMPD was synthesized in a two-step approach by using [1-$^{13}$C]ethyl pyruvate as a precursor. The pH sensitivity of the $^{13}$C resonances in the physiological pH range originates from the C$_5$-carboxyl-group with pK$_{a2}$ = 6.55. **b** $^{13}$C spectra from a titration series of Z-OMPD show strong and weak pH sensitivity for the C$_5$- and C$_1$-resonance, respectively, relative to [$^{13}$C]urea as a nonshifting reference. **c** Fitting of chemical shift changes of Z-OMPD as a function of pH relative to [$^{13}$C]urea with a scaled logistic function yields pH sensor calibration curves. **d** Comparison of T$_1$ relaxation time curves of $^{13}$C-labelled hyperpolarized in vivo pH sensors shows $^{13}$C-labels of undeuterated Z-OMPD (OMPD$_1$; OMPD$_5$) to exhibit superior hyperpolarized signal lifetime compared to deuterated zymonic acid (ZA$_{d,1}$; ZA$_{d,5}$). Relaxation curves are measured in D$_2$O at 1 T. **e** T$_1$ relaxation time constant of the $^{13}$C-labels of Z-OMPD at different field strengths (1 T and 7 T) and solvents (D$_2$O and blood), as well as measured in vivo at 7 T.

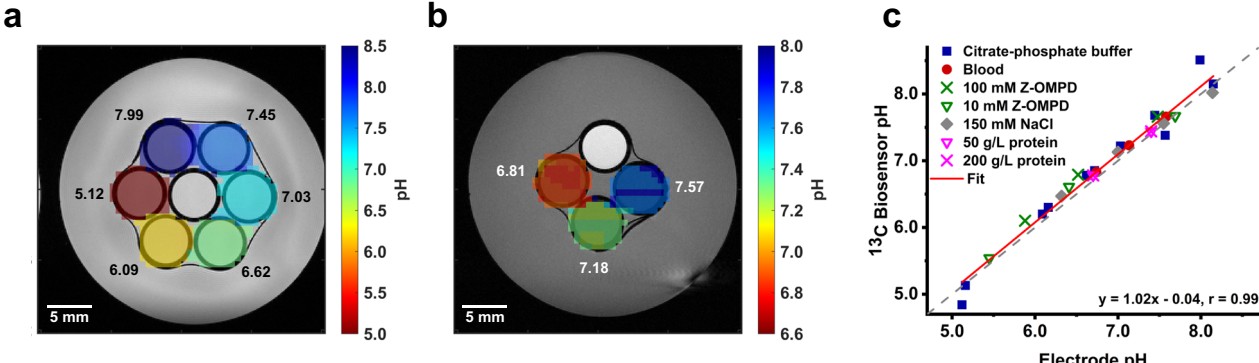

**Fig. 2 | In vitro pH imaging of Z-OMPD in buffer and blood phantoms. a** pH map generated from a CSI acquisition of 6 mM Z-OMPD and 10 mM [$^{13}$C]urea in six citrate-phosphate buffer phantoms of different pH values, overlaid with a corresponding T$_2$-weighted image. Black numbers indicate reference phantom pH values measured using a conventional pH electrode. **b** pH image of three human blood phantoms of different pH values after injection of 11 mM hyperpolarized Z-OMPD and 19 mM [$^{13}$C] urea, overlaid with a T$_1$-weighted image. White numbers display phantom reference pH values. The white center phantom (**a**) and top phantom (**b**) contain only thermal [$^{13}$C]urea for B$_1$ calibration. **c** pH values measured under various citrate-phosphate buffer conditions (phantom conditions from imaging in **a**), blood (phantom conditions from imaging in **b**), pH sensor-, salt- or protein concentrations using Z-OMPD agree well with measurements using a conventional pH electrode.

## Cytotoxicity assessment and cell uptake

Prior to in vivo application of Z-OMPD, basic information about its toxicology was collected. Stability tests showed Z-OMPD to be stable for 48 h in culture medium, which eliminates the risk of decay product-related side effects (Fig. S5a). Incubation with cells and viability assays with in vivo-relevant Z-OMPD concentrations neither show toxicity directly after incubation (Fig. S5b) nor 48 h after incubation (Fig. S5c). Here, intracellular incorporation within 1 hour is observed (Fig. S6).

However, cellular uptake is marginal on time scales relevant for imaging (< 3% for 1 min incubation time), confirming [1,5-$^{13}$C$_2$]Z-OMPD as an extracellular pH sensor for hyperpolarized $^{13}$C MRI.

## pH imaging in healthy rat kidneys

Validation of in vivo pH imaging was done in healthy rat kidneys, which possess physiological tissue acidification due to the renal filtration process. CSI acquisitions on co-injected [$^{13}$C]urea and Z-OMPD show

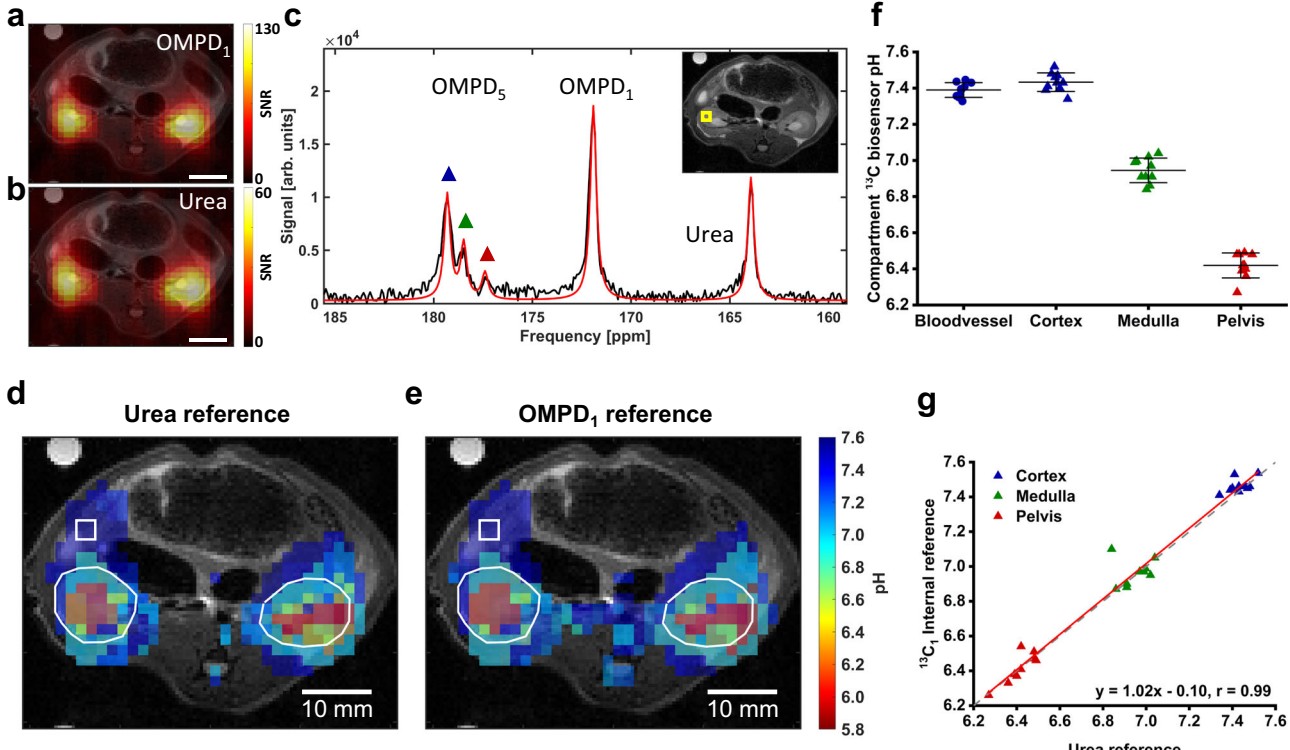

**Fig. 3 | In vivo pH imaging in healthy rat kidneys and internal reference.**
**a**, **b** Signal distribution and signal-to-noise-ratio (SNR) of the OMPD$_1$ (**a**) and [$^{13}$C]urea (**b**) show strong accumulation of both compounds in the kidneys. Scale bars, 10 mm. **c** Single voxel spectrum from inside of a rat kidney. The exact voxel position is indicated in the inset with a yellow square, the T$_{2w}$ image corresponds to the background in **d** and **e**. The C$_5$-resonance of Z-OMPD (OMPD$_5$) splits into three peaks, thereby revealing pH compartments on a sub-voxel level (blue triangle = renal cortex, green = renal medulla, red = renal pelvis). **d** In vivo pH maps of up to three pH compartments of healthy rat kidneys (white ROIs) overlaid on each other and a T$_2$-weighted anatomical image. pH values were calculated using the derived calibration curve and fitting of each C$_5$-resonance position using [$^{13}$C]urea as a reference. The white square indicates the native voxel size. **e** Compartment in vivo pH maps generated as in (**d**) but using the OMPD$_1$ as an intra-molecular internal reference for pH calculations. Compartment maps show good agreement with the urea-referenced version in **d**. **f** A cluster plot shows good consistency for the pH values of the blood pool and the detected renal pH compartments across ten animals. Values are presented as means ± SD ($n = 10$ for each region from independent from independent probe injection experiments). **g** A quantitative comparison of the pH values of detected renal compartments using either [$^{13}$C]urea or the C$_1$-resonance as a reference shows good agreement between the two methods.

stronger signal for Z-OMPD (Fig. 3a) compared to [$^{13}$C]urea (Fig. 3b) with nearly identical signal distributions within the imaged slice. Single voxel spectra of rat kidneys show a cluster of three peaks for the C$_5$-resonance (Fig. 3c), corresponding to different kidney pH compartments. Peaks were fitted voxel-wise and pH values calculated using [$^{13}$C]urea as a reference. The resulting and overlaid compartment maps (Fig. 3d) reveal a global physiologic compartment (Fig. 3c, blue marker), while the first (Fig. 3c, green marker) and the second acidic compartment (Fig. 3c, red marker) are only present in the kidneys and the kidney center, respectively. Based on spatial distribution and pH, compartments can be assigned to the anatomical renal regions of the cortex, the medulla, and the pelvis. Multiple measurements ($n = 10$) on several rats ($n = 7$) show good agreement between animals (Fig. 3f), with a physiological blood pH (pH = 7.39 ± 0.04, $n = 10$) as well as pH values of the cortex (pH = 7.43 ± 0.05, $n = 10$), the medulla (pH = 6.94 ± 0.07, $n = 10$) and the renal pelvis (pH = 6.42 ± 0.07, $n = 10$) agreeing with previous studies[41,50] where pH$_{Cortex}$ = 7.30 – 7.40, pH$_{Medulla}$ = 6.94 – 7.00 and pH$_{Pelvis}$ = 6.30 – 6.55 was reported.

### Internal reference
Co-injection of a reference compound makes probe preparation, handling and hyperpolarization difficult, while also shortening the potential acquisition time window if the reference compound has a shorter T$_1$. Division of the C$_1$-resonance into multiple peaks of different shifts could not be observed in healthy rat kidneys, which suggests the suitability of the C$_1$-resonance as the internal reference for pH calculation. Here, signal intensity and lifetime of the C$_1$-resonance

are superior compared to [$^{13}$C]urea. The pH maps calculated with an internal reference are virtually identical to [$^{13}$C]urea-referenced maps (Fig. 3e), where image features are preserved qualitatively (structural similarity index SSIM$_{cortex}$ = 0.9987; SSIM$_{medulla}$ = 0.9999; SSIM$_{pelvis}$ = 0.9998, all $n = 10$) with minimal uncertainties in pH (root-mean-square-error RMSE$_{cortex}$ = 0.05; RMSE$_{medulla}$ = 0.03; RMSE$_{pelvis}$ = 0.04, all $n = 10$). In addition, the better signal-to-noise ratio for the C$_1$-resonance allows pH calculation in tissue regions where the [$^{13}$C]urea signal is too low. Quantitative comparison of pH compartments determined either by external or internal reference (Fig. 3g) demonstrates no relevant deviations ($r = 0.99$). In conclusion, Z-OMPD allows quantitatively accurate pH imaging without co-injection of an external reference compound.

### Imaging of tumor acidification in subcutaneous EL4 lymphoma in mice
To apply the validated in vivo pH imaging technique in an oncological setting, CSI acquisitions of hyperpolarized [1,5-$^{13}$C$_2$]Z-OMPD in EL4 lymphoma reveal acidification of the tumor mean pH (Fig. 4a). This mean pH is composed of one physiologic compartment, which is globally present in healthy tissue and most of the tumor (Fig. 4b) and a second, acidic pH compartment which is only detectable in tumor subregions (Fig. 4c). Here, mean pH and compartment detection are potentially influenced by tracer extravasation into the extracellular space and signal decay (Fig. S7). Presence of either one or two compartments can be distinguished by either one (Fig. 4d) or two C$_5$-peaks of varying relative intensity (Fig. 4e). pH imaging in multiple

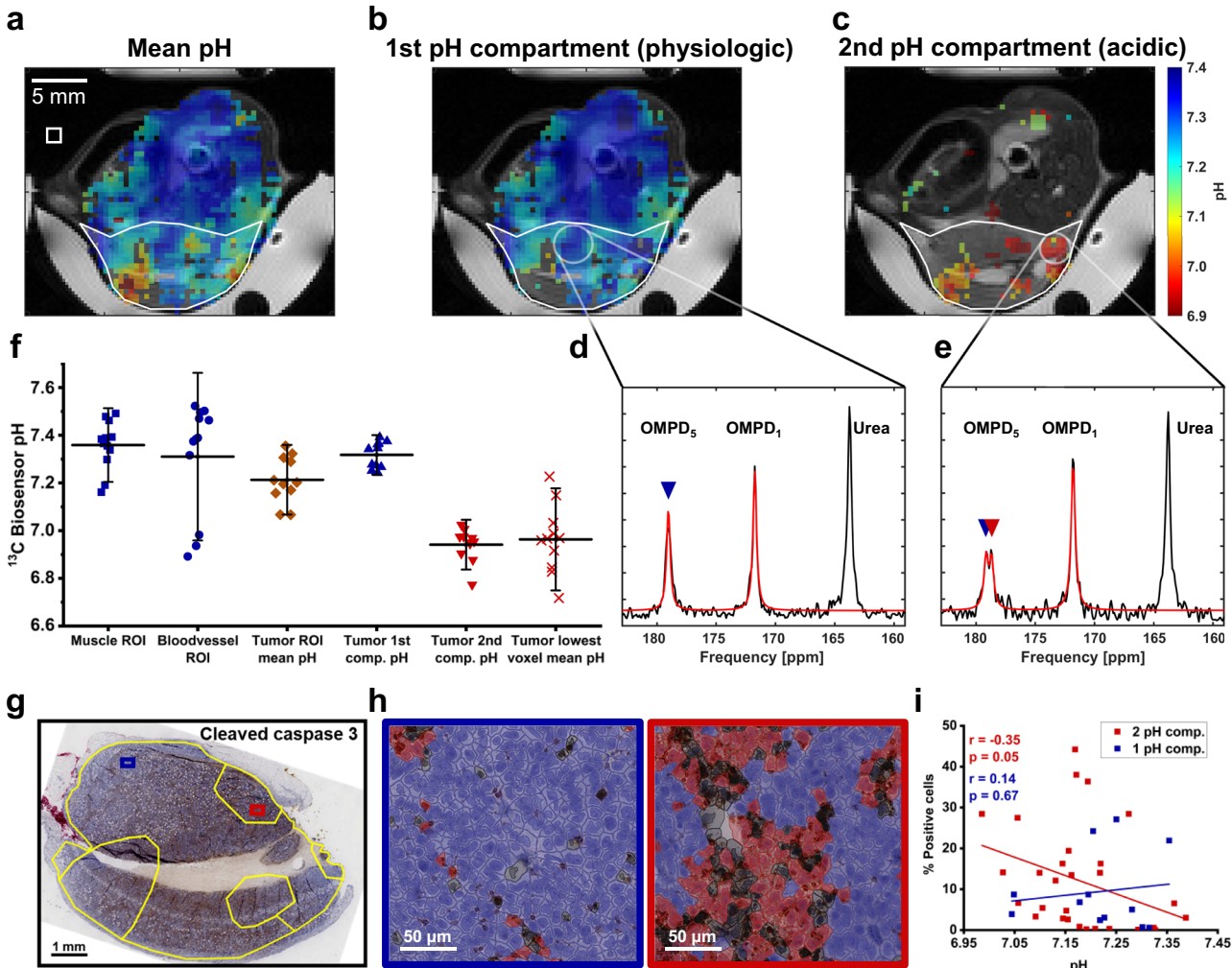

**Fig. 4 | In vivo pH imaging in subcutaneous EL4 lymphoma allows assessment of tumor heterogeneity. a** pH imaging in subcutaneously implanted EL4 lymphoma (white ROI) reveals heterogeneous tumor acidification. The white square indicates the native CSI resolution. **b** The majority of the tumor shows only one pH compartment being close to physiological pH conditions (**d**), indicated by one OMPD-$C_5$-peak (blue triangle, spectrum from white circle ROI in **b**). **c** One or multiple subregions exhibit a second, acidic pH compartment, which is detectable as a second peak for the OMPD-$C_5$-resonance (red triangle) in the spectrum (spectrum from white circle ROI in **c**) (**e**). Tumors generally show acidification of the mean pH (orange diamonds) compared to healthy muscle (blue squares) or blood pH (blue circles) (**f**), with its physiological compartment (blue triangle) being

comparable to muscle pH, while the acidic pH compartment (red triangle) and the lowest single voxel mean pH (red crosses) are acidified by up to 0.6 pH units. Values are presented as means ± SD ($n = 12$ for muscle and blood vessel ROIs, $n = 11$ for tumor ROI-derived values, values are derived from independent probe injection experiments). **g** Histological analysis ($n = 9$ tumors) of the resected tumors confirms heterogeneity. **h** Quantification of immunohistochemical stainings of tumors in **g** are performed by classifying viable tumor cells (necrotic cells in black) into positive (red) and negative cells (blue). **i** Tumor areas bearing an acidic pH compartment tend to show a stronger expression of cleaved caspase 3 with increasing tumor acidification. Correlations for 2- ($r = −0.35$, $p = 0.05$) and 1-pH compartment regions ($r = 0.14$, $p = 0.67$) were assessed by linear regression.

mice indicates a physiologic pH for healthy spine muscle (pH = 7.36 ± 0.10, $n = 12$) and central blood vessels (pH = 7.31 ± 0.23, $n = 12$), with a few subjects showing systemic tissue and blood acidification, potentially due to respiratory acidosis[51,52]. This might occur during anaesthesia which can lead to a severe drop in blood pH[53–55] (Fig. 4f). The physiological pH compartment of EL4 tumors, corresponding to tumor vessels shows no acidification (pH = 7.32 ± 0.06, $n = 11$), while there is also an acidic compartment (pH = 6.94 ± 0.07, $n = 11$), representing extracellular space. The latter causes overall tumor acidification (pH = 7.21 ± 0.10, $n = 11$) and is the dominant compartment in the most acidified regions (pH = 6.96 ± 0.07, $n = 11$). To validate the observed acidification, excised tumors were stained for apoptosis (cleaved caspase 3) (Fig. 4g), pH regulation (carbonic anhydrase IX: CAIX), hypoxia (HIF-1α) and proliferation (Ki-67) (Fig. S8a–c).

MRI-regions of interest (ROIs) of non-acidified (Fig. 4h left) and acidified regions (Fig. 4h right) were transferred to co-registered

immunohistochemistry (IHC) slides to correlate pH and relative IHC marker positivity. Binary threshold-based quantification of positive fractions of viable tumor cells (Fig. S8g) reveals a weak correlation between tumor mean pH and apoptotic fraction (linear regression: $r = −0.35$, $p = 0.05$) for areas showing an acidified pH compartment. No correlation was found for other markers (Fig. S8d-f) or for whole tumor values (Fig. S8h).

## Simultaneous imaging of pH and perfusion in healthy rat kidneys

Beyond sole pH imaging, Z-OMPD can be used for combined imaging of pH and perfusion. For pH imaging with Z-OMPD, pH is retrieved from the $C_5$-resonance, while the $C_1$-resonance is most useful as an internal reference, where the latter requires only moderate SNR. Due to high polarization levels and long in vivo $T_1$ for both resonances, most of the $C_1$-magnetization can be used to image the distribution of Z-OMPD in kidneys within the first 30 seconds after start of injection

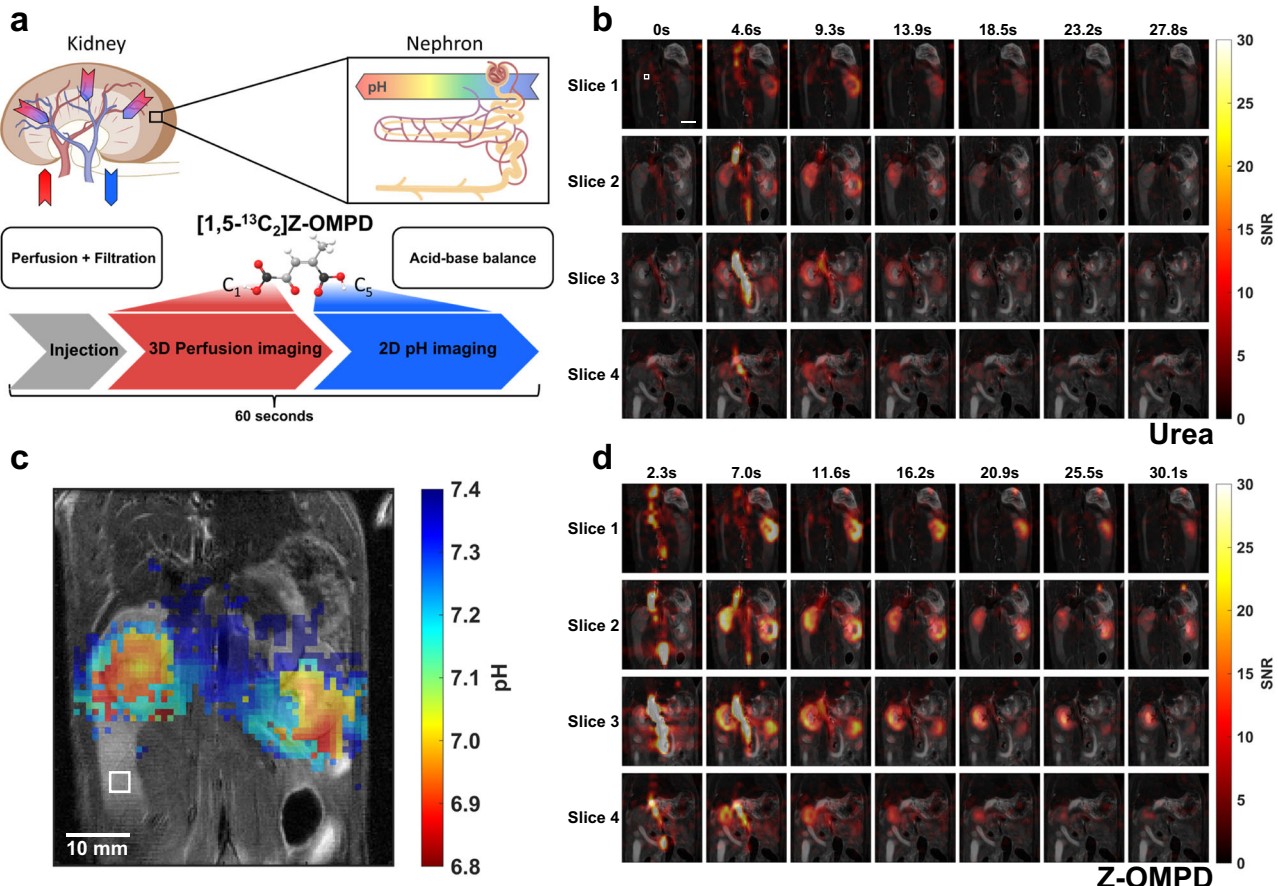

**Fig. 5 | Simultaneous in vivo imaging of renal perfusion, filtration, and acid-base balance. a** Scheme for simultaneous combined imaging of renal kidney perfusion and filtration in 3D and renal pH in 2D within one minute where functional information is selectively obtained from one of the [13]C labels of Z-OMPD. **b** Image time series of co-injected hyperpolarized [13C]urea for a comparison with a standard hyperpolarized perfusion marker covering both healthy rat kidneys. The white square indicates the native voxel size. Scale bar, 10 mm. **c** The following CSI acquisition allows reconstruction of low-noise pH maps where the acidic renal pelvis generates good pH contrast compared to the surrounding renal cortex.

Native voxel sizes for both acquisitions are indicated by white squares. **d** Image time series of the $C_1$-magnetization of hyperpolarized Z-OMPD. The higher SNR compared to [13C]urea together with the high spatial resolution allows better assessment of first-pass perfusion and renal filtration from the cortex (renal periphery) to the pelvis (renal center) when injected at identical concentrations. Image acquisition for both compounds started with start of injection and acquisition of time frames alternated between [13C]urea and Z-OMPD. Hyperpolarized acquisitions are overlaid with an anatomical $T_2$-weighted image. The schematic kidney in **a** was created with BioRender.com.

without affecting its use as an internal frequency reference for later measurements. For initial mapping of perfusion, a narrow-bandwidth frequency-selective 3D balanced steady-state free precession (bSSFP)[56] scan leaves the $C_5$-resonance unaffected, allowing subsequent pH imaging utilizing the remaining $C_1$-magnetization and the full $C_5$-magnetization with a 2D CSI sequence during the following 30 seconds. This forms a hybrid imaging protocol, where renal perfusion, filtration and acid-base balance can be imaged within one minute using a single injection (Fig. 5a).

The alternating acquisition of co-injected [13C]urea (Fig. S9a) and the $C_1$-resonance of Z-OMPD (Fig. S9c) allows direct comparison of perfusion and filtration kinetics in 3D of Z-OMPD (Fig. 5d) to [13C]urea (Fig. 5b) as an established hyperpolarized perfusion marker. Bloch simulations (Fig. S9b, d, e) and phantom experiments confirm that both resonances can be imaged without relevant artifacts or contaminations, regardless of the pH condition (Fig. S10). For in vivo acquisitions, the signal bolus arrives in the central vessel, accumulates in the kidney cortex, and gets filtered to the kidney pelvis (Full slice stack of Fig. 5b, d in Fig. S11a, b). The following CSI-acquisition still reliably captures the acidified renal pelvis indicated by decreased mean pH values in reconstructed pH maps in the respective anatomical region (Fig. 5c).

### Quantitative assessment of renal perfusion and filtration

From 3D ROIs (Fig. 6e, full ROI example in Fig. S12), signal time curves for [13C]urea (Fig. 6a) and Z-OMPD (Fig. 6c) in the renal cortex and pelvis can be obtained. Time curves were corrected with in vivo longitudinal relaxation decay constants for [13C]urea[41] and Z-OMPD (Fig. 1e) and periods of approximately constant cortex signal and linearly increasing medulla signal were used for glomerular filtration rate (GFR)-fitting with Eq. 1 (see Methods, Fig. 6b, d). For renal blood flow (RBF) quantification, additional ROIs (Fig. 6e, red) were placed on central blood vessels to derive input functions for [13C]urea (Fig. 6f) and Z-OMPD (Fig. 6g). Calculation of RBF from input and cortex time curves on central vessels with Eq. 2 (see Methods) yields values between 1 ml/min /ml$_{tissue,Cortex}$ and 7 ml/min /ml$_{tissue,Cortex}$ (Fig. 6h), with strong correlation of [13]C-urea- and Z-OMPD-based measurements ($r = 0.90$, $p = 2 \times 10^{-6}$). Single kidney GFRs show considerable variation (0.5 – 3.5 ml/min) while measurements of [13C]urea and Z-OMPD strongly correlate ($r = 0.88$, $p = 4 \times 10^{-5}$, Fig. 6i). Interestingly, a paired t-test indicates total GFR (tGFR)-values for Z-OMPD (Fig. S13a, tGFR = $4.2 \pm 1.2$ ml/min, $n = 8$) to be systematically higher ($p = 0.002$) compared to [13C]urea (tGFR = $3.3 \pm 0.9$ ml/min, $n = 7$). Here, reabsorption of [13C]urea by urea transporters during the renal filtration process[57] potentially reduces the amount of urea filtrated to the pelvis, resulting

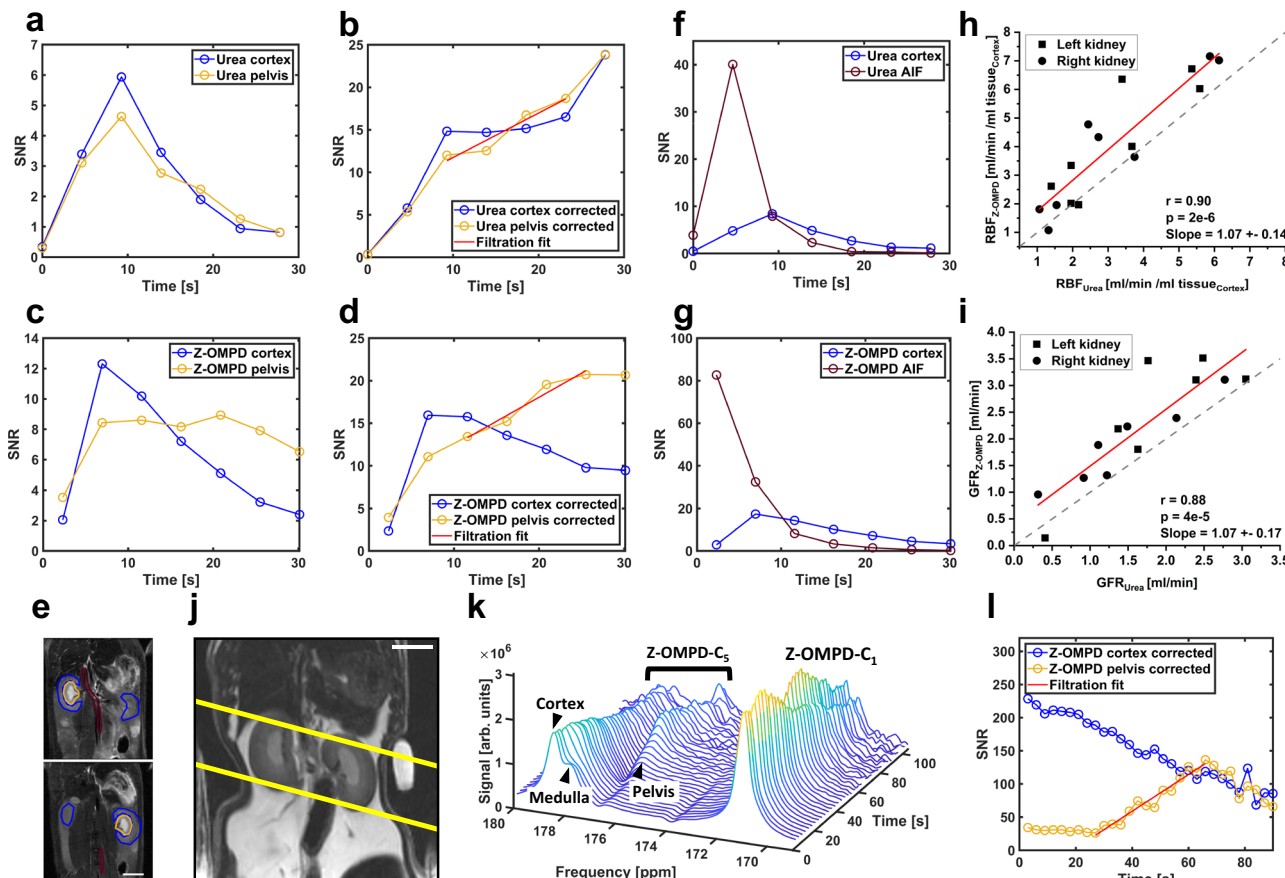

**Fig. 6 | Image- and spectroscopy-based quantification of renal function. a** The accumulation of both [$^{13}$C]urea (**a**) and Z-OMPD (**c**) in the renal cortex (blue curves and ROI) and pelvis (yellow curves and ROI) can be assessed by extraction of 3D-ROI-based (**e**, two example slices shown) signal time curves from the image series in Fig. 5b, c. Scale bars, 10 mm. **b**, **d** $T_1$-decay correction and fitting of time curve sections, where temporal signal evolution is mainly driven by the renal filtration process. Additionally, for renal blood flow, cortex time curves are normalized by image-derived arterial input functions (AIFs, red ROIs in **e**) for [$^{13}$C]urea (**f**) and Z-OMPD (**g**). **h** Single kidney GFR values for [$^{13}$C]urea and Z-OMPD show good agreement, with Z-OMPD showing slightly faster renal clearance. **i** Comparison of renal blood flow values for [$^{13}$C]urea and Z-OMPD also shows good agreement between both perfusion agents with renal blood flow measured by Z-OMPD being systematically higher compared to [$^{13}$C]urea. **j** Axial slice positioning for non-imaging, spectroscopic assessment of renal filtration. Scale bars, 10 mm. **k** Waterfall plot of a dynamic slice-spectroscopy on both kidneys. Due to different pH milieus, the anatomical regions of the kidney, namely the renal cortex, medulla and pelvis are distinguishable as individual peaks for the $C_5$-resonance of Z-OMPD. **l** tGFR values can be calculated from spectroscopic data only by fitting of the time curves of the cortex- and the pelvis peak integrals. Correlations for RBF ($r = 0.90$, $p = 2 \times 10^{-6}$) and GFR ($r = 0.88$, $p = 4 \times 10^{-5}$) between [$^{13}$C]urea- and Z-OMPD-based measurements were assessed by linear regression.

in lower observed filtration rates. Nevertheless, tGFRs agree reasonably well with previous studies[35], where methods using hyperpolarized [$^{13}$C]urea (tGFR = $5.1 \pm 0.9$ ml/min), DCE-MRI (tGFR = $3.5 \pm 1.5$ ml/min and Inulin (GFR = $5.1 \pm 0.9$ ml/min) are reported. Similarly, a paired t-test indicates total RBF (tRBF) (Fig. S13b) to be slightly elevated for Z-OMPD (tRBF = $8.1 \pm 3.9$ ml/min /ml tissue$_{Cortex}$, $n = 9$) compared to [$^{13}$C]urea (tRBF = $6.3 \pm 3.6$ ml/min /ml tissue$_{Cortex}$ n = 8) ($p = 0.03$). Analogous to tGFR, tRBF values agree with previous studies[35] using DCE-MRI (tRBF = $9.5 \pm 3.1$ ml/min /ml tissue) with better agreement for Z-OMPD compared to urea. Alternatively, renal filtration can be probed with the $C_5$-resonance using dynamic slice spectroscopy. For a timeseries of slice spectra (Fig. 6k) covering both kidneys (Fig. 6j), cortex- and pelvis time curves can be extracted due to pH-induced splitting of the $C_5$-resonance. tGFR = 2.9 ml/min, $n = 2$ were fitted from decay-corrected time curves (Fig. 6l) using Eq. 1. Here, overestimation of the cortex signal due to partial overlap with the medulla causes minor underestimation of tGFR compared to the imaging method. As quantitative GFR-, RBF- and pH-values derived from Z-OMPD alone agree with measurements involving co-injected [$^{13}$C]urea, simultaneous imaging of perfusion, filtration and pH is possible without employing co-polarization and co-injection (Fig. S14). Here, faster

acquisition with a slightly modified bSSFP sequence allows higher frame rates for perfusion imaging, together with renal pH contrast maps using the $C_1$-label as an internal reference.

## Multiparametric imaging of hydronephrosis in kidneys caused by pheochromocytoma in *MENX* rats

To demonstrate the value of measuring up to five parameters (RBF, GFR, pH$_{Cortex}$, pH$_{Medulla}$, pH$_{Pelvis}$) within one minute and a single injection, simultaneous imaging of perfusion, filtration and pH by hyperpolarized $^{13}$C-MRI of Z-OMPD was employed to probe the kidney function state in *MENX* rats[58,59]. These develop bilateral adrenal medullary tumors (frequency 100%), alias pheochromocytoma, which oversecrete catecholamines and lead to bilateral kidney hydronephrosis, thereby mimicking clinically observed disease aspects[45]. The 3D perfusion imaging block (Fig. 7a) shows fast and strong perfusion of the renal cortex but no transport of Z-OMPD towards the renal pelvis within the 30 s acquisition time window. The resulting measured renal blood flow (tRBF = $7.3 \pm 4.7$ ml/min /ml tissue$_{Cortex}$, $n = 8$ animals, Fig. 7b) shows strong variation between individuals but no relevant difference compared to healthy kidneys ($p = 0.70$). In contrast, glomerular filtration rates are reduced by more than 75% ($p = 4 \times 10^{-5}$)

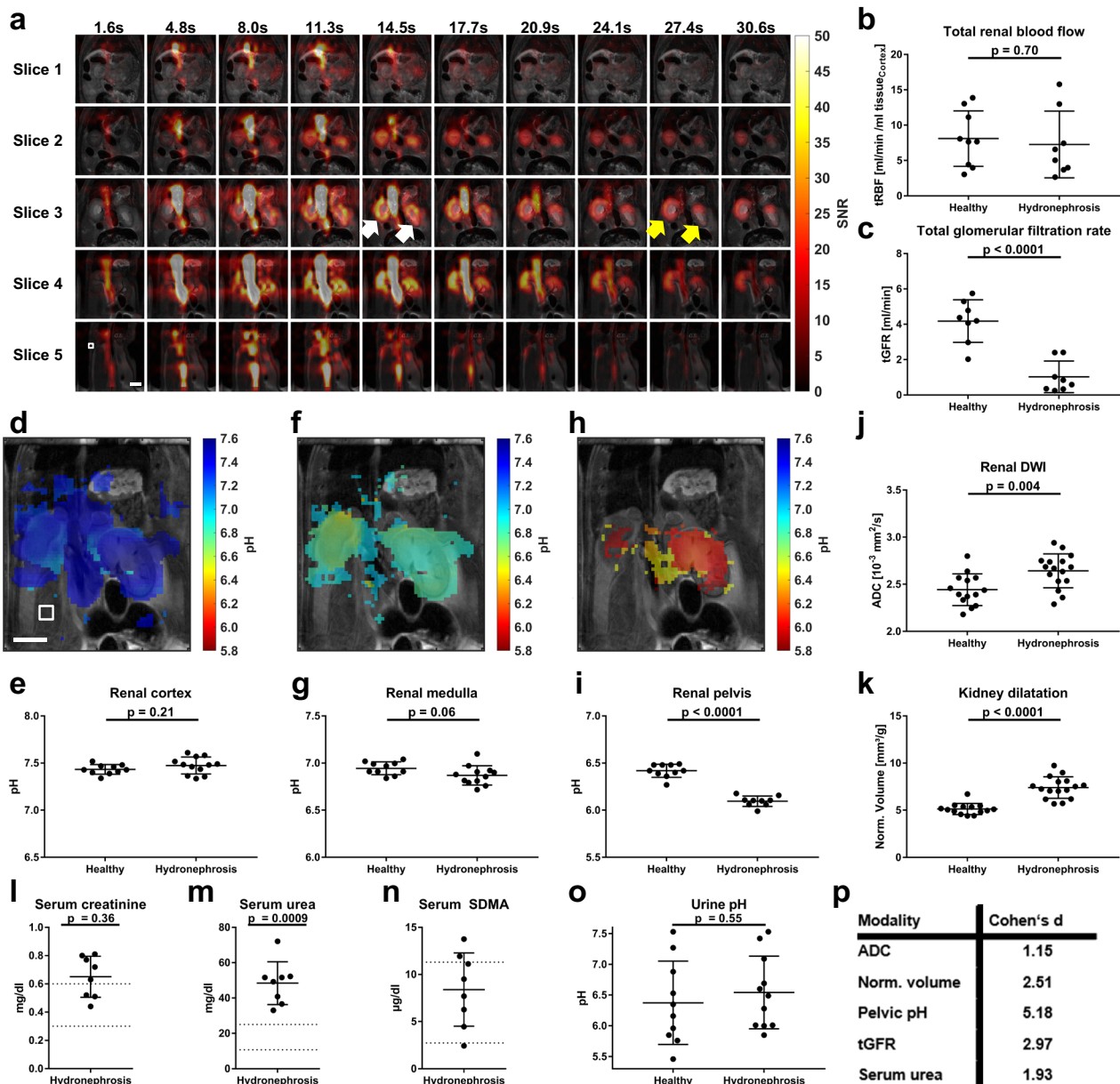

**Fig. 7 | Comparison of kidney parameters in hydronephrosis measured by hyperpolarized ¹³C-MRI using Z-OMPD and clinical standard techniques. a** 3D dynamic perfusion imaging of hydronephrotic kidneys shows poor filtration of Z-OMPD towards the renal pelvis (grey arrows) despite strong cortical perfusion (white arrows). This is quantitatively reflected by normal tRBF (**b,** $n = 9$ healthy and $n = 8$ hydronephrotic individual kidneys) but strongly reduced tGFR (**c,** $n = 8$ healthy and $n = 8$ hydronephrotic individual kidneys). Four-fold interpolated pH compartments for the cortex (**d, e**) and the medulla (**f, g**) appear homogeneous and at physiological pH while the pelvis exhibits strong, pathologic acidification (**h, i**) ($n = 10$ healthy kidney values from Fig. 3f, $n = 12$ individual hydronephrotic kidney cortex and medulla and $n = 9$ pelvis compartments). The left kidney pH compartments in **d**, **f** and **h** exhibit reduced cross-section due to inclined CSI slice placement. Conventional MRI for renal disease status assessment shows a moderate increase **j** in ADC (diffusion-weighted MRI) and a substantial volume (**k**) increase of

hydronephrotic kidneys ($T_2$-weighted MRI, **j**, **k**: $n = 14$ healthy, $n = 16$ hydronephrotic individual kidneys) compared to healthy controls. Renal biomarkers in blood serum reveal mildly increased creatinine- (**l**) and strongly elevated urea levels (**m**), while SDMA values are not clinically evident (**n**) (**l**, **m**, **n**: $n = 8$ individual blood samples). Reference ranges (dashed lines) were obtained from animal suppliers for creatinine and urea[97] or literature for SDMA[98–100]. **o** Urine pH is unobtrusive in hydronephrotic models ($n = 10$ healthy and $n = 11$ individual hydronephrotic urine samples). **p** Comparison of significant parameters using Cohen's d indicates pelvic pH and tGFR measured by hyperpolarized imaging Z-OMPD to be most sensitive to this kidney disease. Scale bars and white squares in **a** and **d** indicate 10 mm and native acquisition resolution respectively. Values in **b**, **c**, **e**, **g**, **i**–**o** are presented as means ± SD. Two-tailed, unpaired Student's $t$ tests were used in **b**, **c**, **e**, **g**, **i**, **j**, **k**, **o**, One-sided Student's t tests against the upper reference interval boundary were used in **l**, **m**.

compared to healthy animals (tGFR = 1.0 ± 0.9 ml/min, $n = 8$, Fig. 7c). Subsequent pH imaging reveals analogous to healthy kidneys three homogeneous pH compartments being assignable to cortex (Fig. 7d), medulla (Fig. 7f) and pelvis (Fig. 7h), with the latter being predominantly present in the pelvic area. All three compartments show large coverage of the kidney due to its anatomical complexity and

limited scan resolution (Fig. S16). Here, pH values of hydronephrotic cortices ($pH_{Cortex} = 7.47 ± 0.09$, $n = 8$ animals Fig. 7e) show no alteration compared to healthy kidneys ($p = 0.21$) and evidence for medullary pH acidification ($pH_{Medulla} = 6.90 ± 0.10$, $n = 8$ animals, Fig. 7g) is non-significant ($p = 0.06$). Contrary, the third pH compartment shows a strong acidification ($pH_{Pelvis} = 6.10 ± 0.06$, $n = 8$ animals, Fig. 7i) in

comparison to healthy controls ($p = 3 \times 10^{-9}$). Together, these five parameters suggest that the renal medulla and pelvis are most severely affected by the hydronephrotic kidney state. Diffusion weighted-imaging and reveals increased single kidney ADC values (ADC = $2.64 \pm 0.18 \times 10^{-3}$ mm²/s, $n = 16$, Fig. 7j) compared to healthy kidneys (ADC = $2.44 \pm 0.17 \times 10^{-3}$ mm²/s, $n = 14$, $p = 0.004$), which follows trends observed in previous studies[60].

Together these findings are supported by anatomical, $T_2$-weighted-MRI where dilated ureters and pelvis regions are observed (Fig. S15a), leading to an increase in normalized kidney volume (Fig. 7k, $p = 3 \times 10^{-7}$). Histologic analysis of hydronephrotic kidneys confirms the pelvic dilatation (Fig. S15b), together with a multitude of pathological alterations (Fig. S15c) of which the majority is located to the glomeruli and downstream of the filtration process in the renal tubuli, which supports the in vivo findings of reduced filtration and pelvic acidification. Nevertheless, partially preserved kidney function and incomplete ureteral obstruction were validated by mass spectrometry imaging and $^1$H NMR, which confirm that Z-OMPD is indeed filtered through the kidney with slower excretion dynamics (Fig. S15d) and can be found in the urine 80 minutes after injection (Fig. S15e), allowing to exclude full kidney failure, in line with the measured tGFR > 0. Further assessment of standard renal serum parameters reveals neither creatinine ($c = 0.65 \pm 0.15$ mg/dl, $p = 0.36$, Fig. 7l) nor SDMA ($c = 8.39 \pm 3.89$ µg/dl, Fig. 7n) to be elevated and only urea ($c = 48.42 \pm 12.13$ mg/dl, $p = 0.0009$, Fig. 7m), which by itself is only a limited indicator for safe diagnosis of kidney disease[61,62], exhibits strongly increased serum levels. Urine shows no relevant susceptibility to the strong acidification in the renal pelvis ($p = 0.55$) despite a slight trend for more alkaline pH values due to obstruction-related inability of acid excretion[63]. Further, all parameters, which were found to be significantly indicative of kidney damage in this hydronephrosis model, are compared regarding their sensitivity using Cohen's d (Fig. 7o). Here, hyperpolarized $^{13}$C-MRI-derived parameters, namely pelvic pH ($d = 5.18$) and tGFR ($d = 2.97$) show extraordinarily high sensitivity to the validated kidney disease in this *MENX* model, which renders simultaneous perfusion- and pH-imaging two-fold superior compared to standard blood counts (serum urea: $d = 1.93$) and conventional anatomical ($d = 2.51$) or diffusion-weighted $^1$H-MRI ($d = 1.15$). In addition, the combined set of five Z-OMPD-derived parameters, of which two are pathologically altered (pH$_{Pelvis}$, tGFR) and three within physiological ranges (tRBF, pH$_{Cortex}$, pH$_{Medulla}$), provides a comprehensive picture of the renal function and disease state for the investigated hydronephrosis model, in good agreement with the histopathological analysis.

## Discussion

While first attempts for clinical translation of pH imaging are emerging[19–23], there is no approved technique to address clinical needs[5,64]. Here, we introduced a hyperpolarized extracellular pH sensor molecule, namely [1,5-$^{13}$C$_2$]Z-OMPD. We developed a new synthesis protocol starting from ethyl pyruvate as precursor and obtained both the unlabelled and $^{13}$C-labelled free acid with high purity. While the synthesis is fast and simple, yielding a highly pure product, the current yield might be improved by milder but more time-consuming synthesis approaches[47]. Once synthesized, Z-OMPD can be stored for months as neither moisture nor heat lead to any notable decay, possibly due to the favorable free energy relative to its tautomers[47]. This is advantageous compared to other in vivo pH sensor molecules like [1,5-$^{13}$C$_2$]zymonic acid which degrade[41]. In addition, the C$_5$-label shows good pH sensitivity in the physiological and pathological pH range while the C$_1$-label only shows a weak sensitivity with opposite sign, potentially due to electron shielding by the C$_2$-keto-group. This pH sensitivity was demonstrated to be unaffected by concentration changes, salt, protein and temperature, thereby allowing accurate pH measurements in vivo. Hyperpolarization by DNP demonstrated

excellent hyperpolarization properties with high polarization levels and $T_1$ relaxation times at various field strengths and in vivo that are among the highest ever reported for $^{13}$C-labelled hyperpolarized agents[48,65,66]. While cytotoxicity assays showed no sign of toxicity up to 15 mM and all animals tolerated the imaging protocols well without adverse events, comprehensive toxicity studies are still required for clinical translation.

For in vivo experiments, a set of spatially highly resolved spectra ($3 \times 3 \times 5$ mm³) were obtained allowing to calculate pH maps for multiple compartments. Using the C$_1$-resonance as internal reference provides a considerable advantage compared to co-injection of [$^{13}$C] urea, as the pH accuracy is equal for both approaches while the in vivo $T_1$ of the C$_1$-resonance is longer compared to [$^{13}$C]urea. This approach is unsuitable for [1,5-$^{13}$C$_2$]zymonic acid[41] or DEMA[65], which exhibit only strongly pH-shifting peaks and therefore require co-injection of [$^{13}$C] urea or [$^{13}$C]tert-butanol as reference, which might exhibit a shorter $T_1$ compared to the sensor molecule and complicate sample preparation[37]. In addition, the internal reference within the same molecule assures both sensor- and reference peak to be in the same extracellular compartment, which might be difficult to achieve for co-injections or protocols using hyperpolarized [$^{13}$C]bicarbonate and [$^{13}$C] CO$_2$[40]. The relatively small chemical shift range for Z-OMPD (-8 ppm) using the internal reference further opens the possibility for EPSI-based imaging protocols.

For pH imaging in tumors, extracellular acidification was traced back to acidified second pH compartments, which could be unravelled spectroscopically. This provides an advantage compared to hyperpolarized [$^{13}$C]bicarbonate, where only a mean pH, weighted by an unknown intracellular fraction, is retrieved ratiometrically. Co-registration of pH maps and histology slides suggests correlation of acidification with the expression of cleaved caspase 3, which is commonly related to apoptosis. Nevertheless, there are recent findings that cancer aggressiveness is also driven by caspase 3 for which an acidified tumor pH might be beneficial[67]. This finding cannot be derived from whole tumor averaging, thereby pronouncing the added value of imaging compared to nonlocalized measurements. Interestingly, no correlation with CAIX expression as a major pH regulator nor proliferative potential was observed, thereby supporting previous findings that acidification is driven by strong lactate excretion for this tumor model[68]. Care has to be taken, as the overlay of MRI and histology suffers from distortion or missing tissue due to histologic processing, which might corrupt ROI-transfer[69]. Further, histology slides only cover 3 µm of tissue, thereby representing only a snapshot of the 3 mm slices covered by MRI.

Beyond imaging of pH, the long $T_1$ and high polarization level enabled the extension of the imaging protocol to dynamically acquire 3D images prior to pH mapping for perfusion and renal filtration assessment. By selective imaging of individual resonances, sufficient signal qualities were achievable for both imaging blocks, rendering the presented protocol to our knowledge the fastest technique to simultaneously assess renal perfusion, filtration, and acid-base balance noninvasively. Both RBF and GFR could be calculated from respective signal time curves and agree with literature[35,70]. However, strong variations between subjects for both GFR and RBF are observed. Here, one limitation is the low number of frames, resulting effectively in only 4–8 time points to capture flow and filtration dynamics. While this might result in some quantification error, large inter-subject variations for GFR (0.03 – 4.3 ml/min) have also been observed using the gold standard inuline[71–73] while radiotracer[74,75], PET[76,77] or sinistrin[78] also show variation of up to a factor of 10 in healthy subjects. This might be explained by varying numbers of nephrons per kidney[79], which itself show single-nephron-GFR variations by up to a factor of 2[71,79]. These can already occur on short time scales caused by blood-pressure-driven autoregulation[80,81] or even simple saline injections[82]. Variations similar to GFR have been reported for RBF[83–85], while sparse temporal

sampling, imaging resolution and bolus dispersion[86] might be the main source of uncertainty in our study. Nevertheless, absolute values and ranges overall agree well with literature, thereby validating our technique. For improvement of accuracy, a protocol without co-injected [13C]urea was shown to allow the acquisition of more frames within the same time, making the RBF and GFR calculation potentially more robust. The assessment of renal function using Z-OMPD also does not require urine-[70] or blood[87] sampling for GFR analysis which typically do not allow to distinguish differences between single kidneys in a subject. One limitation of the current protocol is that perfusion is imaged in 3D while pH is only measured in 2D due to the required high spectral resolution. Here, fast EPSI protocols or phase-sensitive frequency mapping might allow pH mapping in 3D in the future. In addition, Z-OMPD allows for the first time to assess GFR and pH by spectroscopy only, which provides a higher temporal resolution.

The image-based protocol for simultaneous assessment of perfusion and pH was applied to a clinically relevant kidney disease model of hydronephrosis induced by spontaneously occurring bilateral pheochromocytoma (i.e. *MENX* rats). Here, up to five parameters (tRBF, tGFR, $pH_{Cortex}$, $pH_{Medulla}$, $pH_{Pelvis}$) being indicative of renal function status were measured within a single injection in less than a minute, of which three are physiological and two indicate reduced filtration and an acidified renal pelvis which matches histological findings. While these parameters prove to be more sensitive to the present disease features compared to standard blood- or urine testing and routine MRI methods, the entire set of all five parameters could be regarded as a fingerprint of this kidney disease type. Investigation of other kidney disease models will be crucial to determine whether different kidney diseases provide an unambiguous pattern of these parameters. Also, the studied model is limited in studying the sensitivity of hyperpolarized 13C-MRI with Z-OMPD to less severe forms of kidney diseases and its comparison to routine methods. Nevertheless, to our knowledge, no other non-invasive imaging technique allows such a fast and comprehensive assessment of multiple kidney function parameters. Here, in addition to the short scan time being a major advantage, a comprehensive set of kidney parameters is provided simultaneously, which omits the need for the selection of one parameter for function diagnosis. Importantly, simultaneous assessment of multiple parameters is considered to provide more robustness and reliability[88,89] as serial tests might corrupt each other[90]. Established perfusion imaging techniques are unable to assess pH and CEST-based approaches, which try to combine pH and perfusion imaging[7] fail to capture the individual pH compartments and rather return a weighted-pH. Here, the in this study critical pelvic pH compartment would rather contribute with minor weight, rendering the observed acidification difficult to assess. However, further improvement of spatial resolution and spatial coverage using a multi-slice or 3D acquisition technique might help to better spatially disentangle the pH compartments which have their origin in the microscopic heterogeneity and complexity of the kidney.

Apart from nephrology, this multiparametric hyperpolarized imaging protocol could in principle be also applied for simultaneous assessment of perfusion and pH in tumors, where tumor tissue perfusion, vascularization and potentially acidified compartments could be indicative for tumor aggressiveness or treatment response. Prospectively, the presented protocol could be generalized to also assess perfusion and pH in various organs. For clinical translation, imaging at 3 T might benefit from an even longer in vivo $T_1$. Z-OMPD's suitability for clinical translation might be further improved by prolongation of $T_1$ relaxation times via deuteration. Faster polarization might be possible by the synthesis of precursors with unsaturated side-arms, allowing parahydrogen-induced polarization. Regarding urgent clinical need, grading of hydronephrosis, which frequently occurs in neonatal and pediatric patients remains an open challenge[91] for which neither gadolinium- or iodine-containing contrast agents nor 99mTc-MAG3 are

an option and hyperpolarized Z-OMPD could be a promising alternative. In summary, [1,5-13C2]Z-OMPD presents a promising new hyperpolarized extracellular pH sensor, which offers several advantages compared to existing agents.

In particular comparison to zymonic acid with its ring-like geometry, the open-chain structure of Z-OMPD results in several advantages for pH imaging, including longer $T_1$ relaxation time constants, an internal frequency reference (avoiding the co-injection of a reference compound) and high stability in aqueous solution. These molecular properties, together with a dedicated set of tailored MR pulse sequences enabled the simultaneous read-out of three imaging biomarkers (pH, perfusion and renal filtration) within a single imaging session after one injection of Z-OMPD within one minute, which is a unique feature of Z-OMPD as a molecular imaging agent. This versatile, multi-parametric hyperpolarized imaging protocol shows great potential for nephrology and oncology, making it a promising candidate for clinical translation.

## Methods

### Chemicals

13C-labelled [1-13C]ethyl pyruvate as precursor was purchased from Eurisotop (Saint-Aubin, France). [13C]urea was purchased from Merck (Darmstadt, Germany). Sodium bicarbonate and 30% HCl solution were obtained from Sigma-Aldrich (St. Louis, USA). Ethyl acetate and methanol for synthesis were obtained from Merck. HPLC grade solvents were purchased from VWR (Radnor, USA).

### Synthesis of [1,5-13C2]Z-OMPD

A synthesis protocol for 13C-labelled [1,5-13C2]Z-OMPD using [1-13C]ethyl pyruvate as precursor was developed based on a previously reported protocol for unlabelled Z-OMPD starting from methyl pyruvate[47]. 4.0 g [1-13C]ethyl pyruvate was added to 200 ml of a warmed methanol solution containing 9.7 g $Na_2CO_3$ (2.1 eq). The solution was heated to 70 °C and stirred for three hours using a heated water bath and a magnetic stirrer. The bulk sodium bicarbonate was filtered using a frit and the solvent was removed using a rotary evaporator. The intermediate (orange-white solid) was then suspended in 35 ml water and 73 ml ethyl acetate, adjusted to pH 5.5 – 6.0 using 30% HCl and transferred to a separatory funnel. The organic layer was removed, and the aqueous layer was washed twice with ethyl acetate (2 ×13 ml) and extracted. Subsequently, the aqueous layer was further acidified to pH 0 – 1, transferred back to the separatory funnel and 67 ml ethyl acetate and 33 ml 15% NaCl-solution were added. The organic layer now contained the intermediate and the aqueous layer was washed twice with ethyl acetate (2 × 67 ml). The organic fraction was kept, and the solvent was removed using a rotary evaporator. The intermediate was then suspended in 204 ml of 3 M HCl and heated to 65 °C and stirred for three hours using a heated water bath and a magnetic stirrer. The solution was transferred into a separatory funnel containing 200 ml ethyl acetate. The aqueous layer was washed twice with ethyl acetate (2 × 70 ml) and the organic layer containing the product was washed again with 30 ml 15% NaCl-solution. The solution was vacuum filtered, and the solvent was removed using a rotary evaporator. The reaction product was purified by preparative reversed phase high-pressure liquid chromatography (LC-20AP, Shimadzu, Kyoto, Japan) using a preparative Nucleodur C18 HTEC column (Macherey Nagel, Düren, Germany) together with a VarioPrep VP 15/50 Nucleodur C18 HTec precolumn (Macherey Nagel). Purification runs were performed using a linear gradient of 2-20% acetonitrile with a flow rate of 80 ml/min with solvent A being 0.1% Trifluoroacetic acid in milli-Q $H_2O$ (Milli-Q EQ 7000, Merck, Darmstadt, Germany) and solvent B being 0.1% TFA in acetonitrile. Z-OMPD had a retention time of 9 – 11 minutes. The product was shock-frozen in liquid nitrogen and lyophilized *in vacuo* (alpha 1-2 LD plus, Martin Christ, Osterode am Harz, Germany). After storage in the fridge (5 °C) for several hours, a seed was sown (metal

spatula or tiny amount of Z-OMPD from previous stock), resulting in crystallization of the product. Z-OMPD (white powder) was then stored in the fridge. The experimental yield of Z-OMPD was approximately 18% (weight fraction) with a purity of ≥96%, estimated by means of $^1$H NMR spectroscopy (Fig. S1). Synthesis pathways and molecular structures were visualized using ChemDraw 20 (PerkinElmer, Waltham, USA).

## Cytotoxicity assay

An Alamar Blue assay to assess cellular viability was performed. Therefore, $5 \times 10^3$ HeLa cells (ATCC, Manassas, USA), each suspended in 100 μL cell culture medium, were seeded on day 0. To prepare the incubation of the cells with Z-OMPD, aliquots of the medium, containing Z-OMPD (0-15 mM) were prepared and adjusted to pH = 7.9, which was the measured pH of the native cell medium outside of the cell incubator, using 1 M NaOH. After 24 h, on day 1 of the experiment, cell medium was exchanged with the medium containing Z-OMPD at concentrations between 0 and 15 mM to start the incubation. After 24 h incubation, on day 2 of the experiment, the Z-OMPD-containing medium was exchanged with the native medium to finalize the incubation. To read out the cell viability, Alamar Blue HS (Invitrogen, Waltham, Massachusetts) was added (10 % in medium per well) and the cells were incubated for three more hours, in order to enable the resazurin to be reduced to resorufin. The fluorescence signal of resorufin was measured with a spectrofluorophotometer (BioTek, Bad Friedrichshall, Germany). Three technical replications of every condition were performed. The experiment was repeated with the cell viability readout happening on day 4 of the experiment (2 days after the end of Z-OMPD incubation).

## Cell uptake assay

To assess internalization of Z-OMPD into cells, $3 \times 10^6$ EL4 cells (ATCC, Manassas, USA) were incubated with varying concentrations of Z-OMPD (0, 10, 50 mM) in standard DMEM + 10% foetal horse serum for a fixed duration (1, 10, 60 min). Two technical replications of every condition were performed. Following incubation, cells were washed twice with ice-cold PBS, suspended in 500 μL of $D_2O$ and sonicated for 20 minutes. 1D $^1$H NMR spectra (NA = 216) employing water suppression and decoupled $^{13}$C NMR spectra (NA = 1536) of the resulting lysates were acquired on a 11.7 T Bruker AVANCE III 500 NMR spectrometer using Topspin 3.0 (Bruker Biospin, Ettlingen, Germany).

## Characterization of Z-OMPD

Structure, chemical shifts and coupling networks of the synthesized product were confirmed using $^1$H- and $^{13}$C-NMR spectroscopy (Fig. S1) being performed on a 43 MHz Spinsolve Carbon benchtop spectrometer (Magritek, Aachen, Germany; Wellington, New Zealand) of natural abundance Z-OMPD and $^{13}$C-labelled [1,5-$^{13}$C$_2$]Z-OMPD at 1 T (Spinsolve Carbon, Magritek, Germany). The molecular mass of the product was determined by means of mass spectrometry and by electrospray ionization mass spectrometry (ESI MS) using a SQD 3100 Mass Detector (Waters) (Fig. S2). The stability of Z-OMPD in $D_2O$, cell medium, against temperature and sonication was assessed by means of $^1$H NMR-spectroscopy at 1 T (Fig. S4, S5a). pH sensitivity of Z-OMPD and contaminant effects of ionic strength, Z-OMPD concentration and protein concentration (bovine serum albumin) on pH were determined by titration of aqueous solutions containing 25 mM Z-OMPD and 25 mM [$^{13}$C]urea as a reference at 37 °C and acquisition of $^{13}$C NMR spectra on a 7 T small animal preclinical scanner (Agilent Discovery MR901 magnet and gradient system, Bruker AVANCE III HD electronics, Bruker Biospin, Ettlingen, Germany) using 1 M and 10 M NaOH. pH values of phantoms from stock solutions were measured before and after acquisition of each datapoint using a conventional pH electrode (N 6000 A electrode with a ProLab 4000 multiparameter benchtop meter, SI analytics, Mainz, Germany). The mean of these

measurements was taken as reference pH. Temperature sensitivity of $pK_a$ values of Z-OMPD was determined by full titration of 25 mM unlabelled Z-OMPD in $H_2O$ at different temperatures.

## Hyperpolarization and agent preparation

A solution of 16 μl DMSO with 18 mg Z-OMPD (7.1 M) and 0.6 mg OX063 trityl radical (25 mM) (GE Healthcare, Chicago, USA) was mixed for 30 minutes (mostly prepared on the day before and stored as a frozen aliquot until the start of experiment) before being added to a DNP sample cup. For experiments in the kidneys of rats, the amounts were scaled by a factor of 1.5 and for the combined perfusion-pH study, the amounts were scaled by a factor 2.25. For co-hyperpolarization with urea, a 19 μl aliquot of glycerol containing 10 M [$^{13}$C]urea and 30 mM of free radical was first frozen in liquid nitrogen before the Z-OMPD sample was added on top and again frozen in liquid nitrogen. For experiments in rat kidneys, the amounts were scaled by a factor of 1.5 and for the combined perfusion-pH study the amounts were scaled by a factor of 1.3 to equalize final Z-OMPD and urea concentration, being 84 mM for both compounds in the injection solution. The sample was then polarized using a HyperSense dDNP polarizer (Oxford Instruments, Abingdon, UK) at a microwave frequency of 94.231 GHz and 100 mW power for at least 90 minutes. Dissolution was performed using a mixture of 2 ml $D_2O$ containing 80 mM TRIS, 0.1 g/L $Na_2EDTA$ and 227 μL 1 M NaOH in $D_2O$ heated to 180 °C and resulting in injection solutions with pH 7.07 ± 0.58. The final concentrations of Z-OMPD/[$^{13}$C]urea were approximately 56 mM/95 mM for measurements in vitro, in mice and healthy rat kidneys and 84 mM/84 mM for the combined perfusion-pH study. $T_1$ relaxation measurements at 1 T were performed on a Magritek benchtop NMR spectrometer (Software SpinSolve and SpinSolveExpert 1.25) using a pulse-and-acquire sequence with flip angle (FA) = 10° and a repetition time (TR) = 5 s for $D_2O$ as solvent and TR = 2 s for $T_1$ measurements in blood. $T_1$ measurements at 7 T were performed in a preclinical MRI scanner using a pulse-and-acquire sequence with TR = 1 s, FA = 10°. Signal decay curves were corrected for RF depletion and fitted by a three-parameter mono-exponential decay curve. For polarization level calcuations, the solid-state polarization level was determined by referencing the polarization build-up signal magnitude to standard pyruvate sample magnitude of known polarization[92]. The liquid state polarization level was calculated by extrapolation of the signal-to-noise ratio from a decay curve measurement at 1 T back to the time of dissolution and referencing to the SNR from a thermal measurement of the sample at 1 T using a pulse-and-acquire measurement with TR = 600 s, FA = 90°, number of averages (NA) = 400. Thermal reference measurements at 7 T were performed using TR = 700 s, FA = 90°, NA = 65.

## Buffer and human blood phantoms

Aqueous solution buffer phantoms of varying pH were prepared using a 100 mM citric acid / 200 mM disodium phosphate buffer solution. For imaging, six of those phantoms and a 2 mL 0.5 M $^{13}$C-urea phantom for $B_1$-calibration were placed within a hexagonal geometry in a water bath to improve shimming for imaging. 0.2 mL of a hyperpolarized solution containing Z-OMPD and [$^{13}$C]urea were pipetted into each of the 1.8 mL buffer phantom using a multistep pipette (Eppendorf, Hamburg, Germany), resulting in final Z-OMPD and [$^{13}$C]urea concentration of 6 mM and 10 mM respectively. Phantoms were sealed, mixed, mounted in the water bath and placed in the magnet bore prior to acquisition start. Three EDTA-containing blood phantoms (sampled with a S-Monovette®, EDTA K3, Sarstedt, Nümbrecht, Germany) were prepared from fresh human blood and blood pH titrated using 1 M HCl. 0.4 mL of a hyperpolarized Z-OMPD and [$^{13}$C]urea solution were rapidly added to each phantom containing 1.6 mL pH-adjusted blood close to the bore to avoid low-field-related losses of polarization[93] and mixed carefully to prevent bubble formation before being placed in the bore. Final Z-OMPD and [$^{13}$C]urea concentrations in the phantoms

were 11 mM and 19 mM, respectively. Phantom pH values were measured by electrode before and after the MR acquisition. For all phantom measurements, the mean of these measurements was taken as reference pH.

## Animal preparation and handling protocol

All animal experiments were performed in accordance with pertinent laws and regulations and approved by an ethical review board (Regierung von Oberbayern, ROB-55.2-2532.Vet_02-17-177, ROB-55.2-2532.Vet_02-16-117 and ROB-55.2-2532.Vet_02-15-11). All animals were housed in specialized animal housing facilities which are kept at 21-22 °C ambient temperature and 40-60% relative humidity and are operated with a 12-hour light and 12-hour dark cycle with dusk transition periods. All animals had free access to food (Formula 1324, Altromin, Lage, Germany) and autoclaved water. For imaging and tumor implantation, anaesthesia was initiated with 5% isoflurane in 100% oxygen as carrier gas. Second, a tail vain catheter was placed, and animals were positioned on dedicated mouse- or rat beds. Animal body temperature was maintained at 37 – 39 °C by blowing warm air through the magnet bore using a PET Dryer Model B-8 (XPower, San Gabriel, USA). Temperature was monitored using an MR-compatible temperature monitoring system Model 1030 (SA Instruments, Stony Brook, USA). Breathing rate was monitored using an ECG Trigger Unit (Rapid Biomedical, Rimpar, Germany) and kept at 50 – 70 breaths/minute. For hyperpolarized $^{13}$C MRI, solutions were injected via the tail vein catheter. Mice were injected approximately 290 μL of hyperpolarized agent within $t_{inj} = 6.9 \pm 1.4$ s and $23.2 \pm 5.2$ s post start of dissolution. Rats were injected with a dose of 7.5 mL/kg BW of hyperpolarized agent within $t_{inj} = 7.8 \pm 1.9$ s and $17.5 \pm 4.0$ s after start of dissolution. For pH imaging of healthy kidney, 10 healthy female *Wistar* rats (m = 235 ± 16 g, Charles River, Wilmington, USA), 7 – 11 weeks old, were used. For pH imaging of subcutaneous EL4 tumors, 9 female *C57BL/6* mice (m = 20.5 ± 1.1 g, Charles River), 8 weeks old, were injected with 5×10$^6$ cells, suspended in 100 μL PBS, into the right lower back. Tumor imaging was performed on day 8 – 10 following implantation and tumors were removed, cut according to imaging slice position and formalin fixed for histological processing. The maximum tumor size approved by the ethical review board was an average tumor diameter of 15 mm which was not exceeded for all animal experiments. Mice were euthanized under deep isoflurane (5%) anaesthesia and cervical dislocation; rats were euthanized under deep isoflurane (5%) anaesthesia and intraperitoneal injection of 200 mg/kg pentobarbital. For simultaneous imaging of pH and perfusion, nine female *Wistar* rats (m = 247 ± 43 g, Charles River), 9 – 19 weeks old, were used. 5 male and 4 female *MENX* rats[45,58,59], (m = 429 ± 48 g, in-house breed), 8-9 months old, were used as hydronephrosis models.

## Protocols for imaging of pH in phantoms and in vivo

Imaging experiments were performed on a 7 T small animal preclinical MRI scanner (Agilent/GE) MR901 with Bruker Avance III HD electronics (Bruker Biospin, Ettlingen, Germany) and using Paravision 6.0.1. For phantom imaging, a 72 mm dual-tuned $^1$H / $^{13}$C volume resonator for signal transmission and a 30 mm single channel flexible surface coil (Rapid Biomedical, Germany) attached to the bottom of the water bath for signal acquisition was used. For imaging of rats, the same coil as for phantom imaging was used for transmission and a two-channel flexible coil $^{13}$C receive array (Rapid Biomedical, Germany) centred on the back above the rat kidneys was used for signal reception. For imaging of subcutaneous tumors in mice, a 31 mm dual tuned $^1$H / $^{13}$C coil was used for signal transmission and detection. In vivo $T_1$ relaxation time constants for $^{13}$C-labels of Z-OMPD was measured using a slice-selective excitation in axial orientation covering both kidneys with FA = 10°, TR = 3 s, 100 repetitions, slice thickness = 15 mm, excitation bandwidth = 16 kHz, receive bandwidth (BW) = 3.2 kHz, 512 spectral points, total scan time = 300 s. Spectra were zero filled, line broadened by 20 Hz in

MATLAB (The Mathworks Inc., Natick, USA) and fitted with a three parameter mono-exponential decay curve in Origin (OriginLab, Northampton, USA). For hyperpolarized $^{13}$C MRI, for localization and co-registration of kidneys or tumors, $T_2$-weighted $^1$H anatomical images were acquired using RARE with typical parameters: Echo time (TE) = 40 ms, TR = 4000 ms, in-plane resolution = 0.3 mm$^2$, 1 mm slice thickness. pH imaging in phantom, rats, and mice was performed using a free induction decay chemical shift imaging sequence. Acquisition started after phantom placement in the bore or 9 s post end of injection for in vivo experiments. For buffer phantom imaging, CSI used FA = 10°, TR = 83.1 ms, field of view (FOV) = 60 × 36 mm$^2$, slice thickness = 6 mm, resolution 3 × 3 × 6 mm$^3$, BW = 3.2 kHz, 256 spectral points, total scan time = 20 s. For blood phantom imaging, parameters were the same apart from FOV = 64 × 64 mm$^2$, 5 mm slice thickness, resolution 4 × 4 x 5 mm$^3$, total scan time 21.4 s. For pH imaging of kidneys in healthy rats, CSI parameters were FA = 10°, TR = 83.1 ms, FOV = 54 × 42 mm$^2$, matrix size = 18 × 14, slice thickness = 5 mm, resolution 3 × 3 x 5 mm$^3$, BW = 3.2 kHz, spectral points = 256, total scan time = 21.0 s. For high resolution pH imaging in subcutaneous EL4, CSI used FA = 15°, TR = 83.1 ms, FOV = 22.4 × 28 mm$^2$, slice thickness = 3 mm, resolution 1.4 × 1.4 × 3 mm$^3$, BW = 3.2 kHz, 256 spectral points, total scan time = 26.6 s. For high resolution pH imaging in healthy rat kidneys, acquisition parameters were FA = 2.5°, TR = 83.1 ms, FOV = 60 × 54 mm$^2$, slice thickness = 5 mm, resolution 2 × 2 x 5 mm$^3$, BW = 3.2 kHz, 256 spectral points, total scan time = 67 s.

## Imaging protocol for simultaneous imaging of perfusion and pH

To image perfusion, the $C_1$-label of Z-OMPD, exhibiting only a weak pH-sensitive change in chemical shift, was excited using a slightly off-resonant narrow bandwidth excitation, to avoid excitation of the $C_5$-resonance and the co-injected [$^{13}$C]urea peak. Both substances were injected at equal concentrations. Excitation and signal readout were performed with a 3D bSSFP sequence (frequency response profile Fig. S9) starting with start of injection. This acquisition was alternating with a narrow-bandwidth bSSFP acquisition using the same parameters and the excitation and readout frequency being placed on the [$^{13}$C]urea resonance, thereby generating a time series of 3D images for both the $C_1$-label of Z-OMPD and [$^{13}$C]urea to allow a direct comparison of perfusion imaging using Z-OMPD and [$^{13}$C]urea within the same injection. This compound-alternating perfusion imaging acquisition lasting 32 s was followed immediately by a 2D CSI acquisition to image pH. For perfusion imaging of the $C_1$-resonance only, a modified sequence was developed with a larger excitation bandwidth to improve robustness under shim variations. Prior to hyperpolarized acquisitions, transmit $B_1$ was calibrated using a 8 M [$^{13}$C]urea phantom doped with 0.5 mM DOTA placed next to the coil and region of interest and excitation pulses of varying coil power were used to detect the 180° transient. Accurate frequency calibration for the narrow-bandwidth excitation was performed by first measuring the $^{13}$C frequency of the [$^{13}$C]urea phantom or of a Z-OMPD phantom in case of non-alternating acquisitions. This frequency was then fine-tuned to match the frequency within the kidney region due to shim variations between the phantom and the kidney regions. For this purpose, relative frequency differences of the water peak from a $^1$H press acquisition on the phantom and the kidney region were acquired. 3D bSSFP parameters for alternating perfusion imaging were FA = 12°, excitation bandwidth = 180 Hz, TE = 6.4 ms, TR = 12.8 ms, readout bandwidth = 51 kHz, FOV = 60 × 54 x 30 mm$^3$, matrix size = 20 × 18 x 10, resolution 3 mm$^3$ isotropic, number of frames per compound = 7, frame scan time per compound = 2.3 s, temporal resolution 4.6 s for each compound, total scan time 32.4 s. For imaging of the Z-OMPD $C_1$-resonance only without co-injected [13C]urea, the acquisition used FA = 12°, excitation bandwidth = 240 Hz, TE = 5 ms, TR = 10 ms, readout bandwidth = 39 kHz, number of frames = 9, frame scan time = 1.8 s, temporal resolution 3.6 s, total scan time 32.5 s. The CSI acquisition following bSSFP used

FA = 10°, TR = 83.1 ms, FOV = 60 × 54 mm², slice thickness = 5 mm, resolution 3 × 3 x 5 mm³, receive bandwidth = 3.2 kHz, 256 spectral points, total scan time = 29.9 s. Hydronephrosis models were imaged with bSSFP parameters FA = 12°, excitation bandwidth = 240 Hz, TE = 5 ms, TR = 10 ms, readout bandwidth = 32.9 kHz, FOV = 64 × 64 x 40 mm³, matrix size = 16 × 16 x 10, resolution 4 mm³ isotropic, number of frames = 10, frame scan time = 1.61 s, temporal resolution 3.22 s. total scan time 32.2 s and CSI parameters FA = 10°, TR = 83.1 ms, FOV = 64 × 56 mm², slice thickness = 4 mm, resolution 4 mm³ isotropic, receive bandwidth = 3.2 kHz, 256 spectral points, total scan time = 18.6 s.

### Diffusion-weighted imaging

Imaging of diffusivity used a standard Bruker DtiEpi-Sequence with echoplanar imaging readout, 16 $b$-values (10, 20, 40, 60, 80, 100, 200, 300, 400, 500, 600, 700, 800, 1000, 1200, 1500 s/mm²), monopolar diffusion encoding and typical parameters TE = 25 ms, TR = 5000 ms, readout bandwidth = 100 kHz, FOV = 60 ×66 x 30 mm³, resolution 1 mm³ isotropic, 6 repetitions, total scan time 8 min 30 s. ADC values were fitted pixelwise to the signal decay curves with increasing $b$-values using a mono-exponential function with offset.

### pH retrieval and pH map reconstruction

Analysis of spectroscopy data was either performed in MATLAB or MNova (Mestrelab, Santiago de Compostela, Spain). Data analysis of imaging data was performed in MATLAB. Unless stated differently, spectra were zero-filled by a factor of two in spectral dimension and interpolated by a factor of three in spatial dimensions. For fitting of pH maps, [¹³C]urea and up to three pH compartment Z-OMPD peaks were detected using a minimum SNR threshold of 3. Peaks were fitted using least-squares with a linear combination of Lorentzian curves. For pH measurement relative to [¹³C]urea as external reference, the difference between each compartmental $^{13}C_5$-resonance position and the urea peak position was taken. For pH map reconstruction using the Z-OMPD $C_1$-resonance as internal reference, the difference of each compartmental $^{13}C_5$-resonance relative to the dominant $^{13}C_1$-resonance was taken. For generation of in vitro pH maps, spectra were zero-filled by a factor of two in spectral dimension and interpolated by a factor of 4 in spatial dimensions. Z-OMPD is a pH sensor operating in fast chemical exchange regime, thus the chemical shifts of the $^{13}C$-labelled positions can be modelled as a function of pH by a scaled logistic function Z-OMPD$_i$ (pH) = Z-OMPD$_{i,min}$ + $\delta_i$ / (1 + 10$^{(pKa-pH)}$) with the same acid dissociation constant pK$_a$ for the titration curves of both $^{13}C$ labels, with both calibration curves being fully described by Z-OMPD$_{5,min}$ (12.4801), Z-OMPD$_{1,min}$ (8.4919), $\delta_5$ (3.2975), $\delta_1$ (0.4644), and pK$_a$ (6.5450). Some voxels, in particular kidney voxels, contained more than one $^{13}C_5$-resonance of Z-OMPD. Those were assigned to sub-voxel pH compartments. Unless stated differently, detected pH compartments were weighted according to their respective peak intensity and averaged to calculate a mean pH map. The mean pH values for each voxel were colour-coded and the respective compartment pH maps were overlaid on an anatomical $T_2$-weighted RARE image for graphical representation. Statistical analysis was performed using GraphPad Prism 7 (GraphPad Software, San Diego, USA). Schematic image design was supported by BioRender (BioRender, Toronto, Canada).

### Quantitative analysis of immunohistochemical stainings

Following fixation for at least 48 h in formalin, EL4 tumors were immunohistochemically stained for CAIX, Ki-67, cleaved caspase 3 and HIF-1α. Scanned slides were analyzed in QuPath 0.3.2[94]. To derive fractions of positive stained cells for each slide, staining vectors for diaminobenzidine and haematoxylin were estimated for each slide individually by selection of strongly positive and negative tissue regions and fine-tuned using the auto-detection algorithm. MRI tumor images were manually co-registered with histology slides by an affine transformation using the Image Alignment Extension to project MRI-based ROIs onto the histology slides. In a first step, cell detection and classification were performed using the built-in positive cell detection algorithm with a binary threshold. Detection and classification parameters for each IHC staining are listed in Fig. S8d. In a second step, a random trees classifier was trained for each staining with 120 necrotic and 120 viable tumor areas taking the nucleus features of the haematoxylin channel as measures (Fig. S8d), to further distinguish between viable and necrotic tumor areas. For correlation assessments, only viable tumor areas were considered.

### Calculation of glomerular filtration rates and renal blood flow

Analysis of bSSFP acquisitions to assess perfusion and glomerular filtration was performed in MATLAB. For glomerular filtration rates, two 3D regions, one covering the renal cortex and one the medulla and pelvis were drawn slice-wise based on anatomical $T_2$-weighted images. Time curves for both [¹³C]urea and Z-OMPD for both regions were then extracted, decay corrected using in vivo $T_1$-values for Z-OMPD measured in this work and for [¹³C]urea determined in previous works[41]. A two-compartment model with unidirectional flow according to Baumann and Rudin[35,95] was used for modelling according to:

$$\frac{dC_{Medulla+Pelvis}(t)}{dt} = k_{cl}C_{Cortex}(t) \tag{1}$$

where $C_{Medulla+Pelvis}(t)$ and $C_{Cortex}(t)$ are the signal time curves for the cortex and the combined medulla and pelvis region derived from the 3D images and $k_{cl}$ is the clearance rate. For modelling, the equation was linearized and $k_{cl}$ calculated from the linear fit slope of the medulla signal and the average cortex signal at late time points in the dynamic bSSFP acquisitions. To obtain single kidney GFRs in units of ml/min, the resulting clearance rate unit was converted from 1/s to 1/min and multiplied with the total kidney volume measured from $T_2$-weighted ¹H images. Total glomerular filtration rates (tGFR) where then obtained from the sum of both GFRs for the left and the right kidney. For renal blood flow, 3D regions on a central blood vessel (vena cava/aorta) and the renal cortex were drawn and time curves for [¹³C]urea and Z-OMPD extracted to calculate the renal blood flow for individual kidneys according to

$$RBF = \frac{\sum AUC_{Cortex}}{\Delta t \cdot \sum AIF} \tag{2}$$

With AUC$_{Cortex}$ being the summed signal of the renal cortex ROI, $AIF$ being the signal time curve of the central blood vessel and $\Delta t$ the delay between two image frames. Total renal blood flow (tRBF) was then obtained from the sum of both RBFs for the left and the right kidney. For calculation of glomerular filtration rates based on slice spectroscopy data, pH-shifted peaks were assigned to the anatomical regions of the kidneys, signal time curves for cortex and pelvis extracted and used for calculation of tGFR-values.

### Mass spectrometry imaging

Rat kidneys were excised 80 minutes following [1,5-$^{13}C_2$]Z-OMPD injections and immediately dry ice frozen. For mass spectrometry imaging, 10 μm thick sections of the frozen samples were prepared using a Leica CM1950 cryostat and thaw-mounted onto Superfrost glass slides. Spatial metabolomics experiments to visualise [1,5-13C2]Z-OMPD alongside the endogenous tissue metabolome were performed using desorption electrospray ionization mass spectrometry imaging (DESI-MSI) using a Q-Exactive Plus mass spectrometer (Thermo Scientific, Bremen, Germany) operated in negative ion mode. Images were recorded with a mass range of 100-500 Da, mass resolution of 70,000 (at m/z 200) and injection time of 250 ms at a lateral

resolution of 75 µm. Electrospray was composed of 95% methanol at a flow rate of 1.25 µL/min with a spray voltage of 4.5 kV and nebulising gas pressure of 7 bar ($N_2$ purity N5.0). Other settings were: 320 °C capillary temperature, S-Lens RF setting of 50, sprayer-to-sample distance of 1.5 mm, sprayer-to-inlet distance of 6 mm, spray angle at 75° and collection angle at 10°.

## Reporting summary

Further information on research design is available in the Nature Portfolio Reporting Summary linked to this article.

## Data availability

Raw data from NMR spectroscopy, magnetic resonance imaging, mass spectrometry imaging and histology generated in this study have been deposited to mediaTUM: https://mediatum.ub.tum.de/1715708 (https://doi.org/10.14459/2023mp1715708)[96]. Source data are provided with this paper. All information needed to reproduce this study can be found in the manuscript, figures and supplementary information.

## Code availability

The code used to process the data within this article have been deposited to mediaTUM: https://mediatum.ub.tum.de/1715708 (https://doi.org/10.14459/2023mp1715708)[96].

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

## Acknowledgements

We acknowledge helpful discussions with Prof. Dr. Hannes Notni, Dr. Andro Rios, Dr. Beatrice Stephanie Ludwig, Dr. Lisa Russelli and Dr. Frits van Heijster for Z-OMPD synthesis. We thank Prof. Dr. Susanne Kossatz, Nia Nguyen and Hannes Rolbieski for help with cytotoxicity assays, Prof. Dr. Wolfgang Eisenreich and Christine Schwarz for support with high-field NMR spectroscopy, Natalie Röder for help with animal experiments, Marion Mielke, Ulrike Mühlthaler and Olga Seelbach for processing of histological stainings, Dr. Tanja Groll and Dr. Simone Ballke for evaluation of histological stainings. We acknowledge financial support from the Deutsche Forschungsgemeinschaft (DFG, German Research Foundation, Sonderforschungsbereich (SFB) 824, grant number 391523415) to NSP, KS and FS, the Young Academy of the Bavarian Academy of Sciences and Humanities to FS and the European Union's Horizon 2020 research and innovation program under grant agreement No 820374 to FS. Research reported in this publication was supported by the German Federal Ministry of Education and Research (BMBF) in the funding program "Quantum Technologies – from Basic Research to Market" under the project "QuE-MRT" (contract number: 13N16450) to FS.

## Author contributions

P.W. synthesized Z-OMPD, M.G. and P.W. purified the product. M.G. and J.G.S. developed MRI sequences and imaging protocols. M.G., P.W., J.G.S., S.S. and N.Se. conducted the MRI experiments, M.G. and P.W. conducted the NMR experiments. P.W. and S.S. performed the cytotoxicity assay, M.G., S.S. and N.Se. performed animal handling, tumor cell growth and implantation, M.G. and T.M. performed the histologic analysis in QuPath, T.M. and K.S. supervised and validated all histologic analysis. M.G. and P.W. evaluated and analysed imaging and spectroscopy data, S.G., H.M. and N.S.P. provided hydronephrosis models and supported related experiments and analysis. M.P., D.W. and N.St. performed mass spectrometry imaging experiments and analysis. M.G., P.W. and F.S. wrote the paper and all authors reviewed the manuscript, M.G., P.W., J.G.S., C.H. and F.S. designed the research, and M.G. and F. S. devised the study.

## Funding

## Competing interests

The authors declare no competing interests.
