## [Peer Review File · Nature Communications]

Reviewers' comments:

Reviewer #1 (Remarks to the Author):

This paper describes the use of a new ^{13}C hyperpolarized imaging agent ($[1,5-^{13}\text{C}_2]\text{Z- OMPD}$) which possess long T_1 s and a high pH sensitivity for the splitting between the C1 and C5-labelled chemical shifts to allow pH measurements of the C1-labelled shift and an imaging protocol which was developed to quantify perfusion, filtration and pH by combining this with urea. The compound and protocol was tested in healthy rat kidneys and in a subcutaneous lymphoma mouse model. This appears to potentially be a novel imaging agent that was developed with well tuned physical properties for the imaging diagnostic imagined and this is also a worthwhile diagnostic. With that said, the paper could be improved, the following points are made:

- What initiated the idea to synthesize Z-OMPD as a hyperpolarized agent for pH imaging? If it is so similar to one that was previously reported in reference 35, this should be a society journal manuscript so please provide a little more than in the first sentence in pH sensitivity section of Results "Based on Z-OMPD's structural similarity to previously reported pH sensors". Some mention is made in the discussion section, but this needs to be clear at the start of the results section. Perhaps it is also that this synthesis is particularly novel as a rationale? Please highlight why this work is a significant advance compared to established literature.

Introduction

-paragraph 1, "In clinical routine, kidney function is probed by perfusion imaging using arterial spin labelling"? This is a research tool but not in standard clinical routine. Kidney function is routinely probed by iodinated contrast multi-phase CT and MAG3 nuclear scintigraphy using MAG3 as imaging agent, please rephrase.

-paragraph 1, "MRI- and CT-based contrast agents can induce nephropathies,"gadolinium MR contrast doesn't really induce a nephropathy but instead there is a risk of "nephrogenic systemic fibrosis" which involves fibrosis of skin, eyes, joints and internal organs for patients who have impaired kidney function. Please rephrase.

-The first few sections in Results are quite choppy and partially read like methods sections instead, which they shouldn't. They should smoothly describe the premise of the experiments performed and then report sequentially the observations that can be made based on the figures and tables. Please improve this. For example, for paragraph 1, sentence 1 should mention what initiated the idea to synthesize Z-OMPD as a hyperpolarized agent for pH imaging then describe the details of the synthesis yields and isotopic enrichment levels.

-Results, pH imaging in healthy rat kidneys section, When there are literature values for the pH of compartments mentioned in results, they should be explicitly mentioned, i.e. cortex was previously pH = xxx, renal pelvis pH = yy instead of simply that your measurements "agree with previous studies"

-Figure 1, The legend and caption for figure 1d are a little confusing, please mention what ZAd,1 and ZAd,5 are compared to OMPD1 and OMPD5 in either the figure caption which is presumably zymonic acid and Z-OMPD. It would be nice to introduce zymonic acid's structure and describe the results of this compared to Z-ompd in paragraph 2 of results where the invitro pH sensitivity is described. This is also related to the choppiness of the Results section.

-Figure 2 (labeled as a second Figure 1), please mention the buffer and concentration in the legend for Figure 2c instead of just stating "buffer". For the others, the concentrations and items are more explicitly mentioned. Also, is this taken to be at a different concentration in Z-OMPD than 10 mM or 100 mM which are also listed?

-Figure 3, This figure could use the T2w image showing the kidneys for panel d and e, please display it. Particularly since it is hard to see where the cortex, medulla and pelvis are without this.

-It is a bit weird structurally to have a section for pH of healthy rat kidneys then another for pH of tumors and then go back to kidneys and have a section for pH and perfusion of healthy rat kidneys, why wouldn't the pH and the perfusion of kidneys sections be back to back?

-Figure 4, Is it not clear from the images in panels a, c that the ROI is in a muscle compartment, is it possible that it is instead an ROI in the peritoneal cavity or intestines or other compartment? It seems a little odd if it is muscle firstly that there is much contrast in muscle and also that it is 0.2 units lower in pH than the muscle for the other animals? The observation could be explained by the ROI not being muscle but something else instead as there is probably a lot of free probe outside the muscle in the peritoneal cavity.

-Figure 4, Are there differences in the time course for observing this second pH compartment for the lymphoma tumors or do both compartments come up at similar timings after injecting Z-OMPD? This might be the case, it would be nice to understand if it is or not.

- The time courses in figure 5d and 6 a-d seem a little short for measuring clearance of probe based on the prior literature using CT and MAG3 scintigraphy as this should be measured over minutes instead of only 30 s. Granted that the polarization persists for at $t_{1/2}$ of 30 s in vivo based on Figure 1e, but there is also not a good validation of the finding.

-Results for fig 5,6, It would be nicer to mention the measured GFR literature numbers at least in the discussion and describe how similar or different they are for your experiments instead of generally mentioning they agree but not listing numbers.

Reviewer #2 (Remarks to the Author):

The manuscript by Grashei et al. "Simultaneous Magnetic Resonance Imaging of pH, Perfusion and Renal Filtration using Hyperpolarized ^{13}C -labelled Z-OMPD" describes a novel pH imaging biomarker based on hyperpolarized ^{13}C MRS. The biomarker is novel, however the use case of renal cancer, is less so and the additional GFR and perfusion estimation is not utilized in any significant way. pH is important, GFR is important, and perfusion is important, but the paper fails in describing why the simultaneous acquisition is important?

Many different perfusions and GFR estimators exist, yet few enter clinical use and the fact that the data is acquired at the same time, has little real impact in the clinical setting. eGFR is still today the most used GFR measure. Renal MRI perfusion is not a very used clinical tool, although it has great potential.

In summary the most noteworthy results are the novel pH sensor, it has as described in this manuscript little impact, as the combination adds little value, and the data here does not show any benefit over any other pH measurements, in fact the compartment pH data (figure 4) suggest low specificity. The authors claim significant downfalls of CEST and other techniques and thus it would be beneficial to demonstrate this or demonstrate that the unique simultaneous nature of the RBF/GFR/pH measurement has a benefit over existing methods or yields a completely new information not before obtainable.

I recommend a major revision with a requirement to add additional data on the pH biomarker (comparison to alternative techniques) or the combination of pH/GFR/perfusion.

Reviewer #3 (Remarks to the Author):

Grashei et al. present the manuscript entitled "Simultaneous Magnetic Resonance Imaging of pH, Perfusion and Renal Filtration using Hyperpolarized ¹³C-labeled Z-OMPD. Overall, this is an exciting, rigorous, and well written manuscript presenting data which will be of significant interest to the community of researchers engaged in the development of pH imaging methods and hyperpolarized probes. The conclusions are clearly supported by the data presented. No major flaws in data analysis or presentation were noticed.

Some minor edits are suggested:

1. For figure 3c, where is this single voxel location on the rat kidney?
2. The discussion is a "wall of text." Would recommend to reformat a bit to break down into three paragraphs or so.
3. Page 1 - L14. T1 at what magnetic field strength?
4. Check the manuscript throughout to use consistent nomenclature throughout. In some cases [¹³C] is used, in other cases the brackets are not present.
5. Page 2 L4. Add following reference with 31. It was the first patient study.
Liu, X. et al. Development of specialized magnetic resonance acquisition techniques for human hyperpolarized [¹³C,¹⁵N₂]urea + [1-¹³C]pyruvate simultaneous perfusion and metabolic imaging. *Magn. Reson. Med.* 88, 1039–1054 (2022).
6. The abbreviation definitions, like Z-OMP, IHC, RBF or others are also in Methods, they may need to move to the first time they appear.
7. Replace heading of "pH imaging in tumors in mice" to something more specific to detail what is entailed in the section.
8. Page 8 L2. Is "quite squares" – I assume this means white squares?
9. Page 9 L43. "no sign of toxicity" up to what the maximum concentration of probe is.

Point-by-point response to reviewers' comments

We want to address the comments made by the reviewer in this detailed response letter and we have modified our manuscript based upon the reviewers' suggestions. Here, changes in the manuscript with respect to the initial submitted version are marked in yellow. Reviewer comments are written in italic letters and our respective responses are listed below with indentation. All references included in excerpts from the manuscript are indexed according to the manuscript, all others according to this letter.

Reviewer #1 (Remarks to the Author):

This paper describes the use of a new ^{13}C hyperpolarized imaging agent ($[1,5-^{13}\text{C}_2]\text{Z- OMPD}$) which possess long T_1 s and a high pH sensitivity for the splitting between the C_1 and C_5 -labelled chemical shifts to allow pH measurements of the C_1 -labelled shift and an imaging protocol which was developed to quantify perfusion, filtration and pH by combining this with urea. The compound and protocol was tested in healthy rat kidneys and in a subcutaneous lymphoma mouse model. This appears to potentially be a novel imaging agent that was developed with well tuned physical properties for the imaging diagnostic imagined and this is also a worthwhile diagnostic. With that said, the paper could be improved, the following points are made:

We appreciate that the reviewer acknowledges the extraordinary properties and suitability of Z-OMPDP for diagnostic imaging. In this context, we would like to clarify a few points:

First, the C_5 -label and its chemical shift provides the sensitivity for pH measurements in vivo, not the C_1 -label. The latter serves, as a chemical shift reference as it is not markedly shifting within the relevant physiologic pH range. The combination of one strongly and one almost non-shifting ^{13}C -resonance for varying pH is a unique feature of Z-OMPDP and not being reported for any other molecule of comparable size and relevant for medical imaging.

Second, we show that ^{13}C -labelled $[1,5-^{13}\text{C}_2]\text{Z-OMPDP}$ by itself is sufficient to measure GFR, RBF and pH (Fig. S14) which does not require to coinject urea. The co-injection of urea for the experiments shown in Fig. 5, 6 and the combined acquisition was performed to benchmark the capability of Z-OMPDP vs. urea to measure perfusion and filtration. In this comparison, Z-OMPDP shows a higher SNR compared to urea which has been established previously as a hyperpolarized agent for perfusion and kidney filtration¹. Furthermore, this comparison is relevant due to ongoing effort to introduce hyperpolarized urea as a co-injected perfusion agent for clinical studies².

What initiated the idea to synthesize Z-OMPDP as a hyperpolarized agent for pH imaging? If it is so similar to one that was previously reported in reference 35, this should be a society journal manuscript so please provide a little more than in the first sentence in pH sensitivity section of Results "Based on Z-OMPDP's structural similarity to previously reported pH sensors". Some mention is made in the discussion section, but this needs to be clear at the start of the results section. Perhaps it is also that this synthesis is particularly novel as a rationale? Please highlight why this work is a significant advance compared to established literature.

The reviewer raises the important point that the differences and advantages of Z-OMPDP compared to zymonic acid should be highlighted more. In the following we will provide detailed insight into the idea why we synthesized ^{13}C -labelled Z-OMPDP, followed by a list of differences between zymonic acid (ZA) and Z-OMPDP, and lastly addressing the novelty of the synthesis process itself.

1. What initiated the idea to synthesize Z-OMPD?

“When designing new probes and predicting their chemical and biochemical behavior, the success of [1-¹³C]pyruvic acid provides an important exemplar.” (P5, L8-10)³ is a quote from a landmark paper on the design and introduction of new hyperpolarized probes for hyperpolarized ¹³C-MRI by Keshari et al. Hyperpolarized [1-¹³C]pyruvate is a molecule that is now widely used in oncological clinical studies in various imaging centers around the world (<https://clinicaltrials.gov/ct2/results?cond=hyperpolarized+pyruvate>) Here, key aspects for its success are that pyruvate provides important image-based information about the metabolic behavior in tumors, its favorable hyperpolarization properties, given by a long T₁ time and a good polarization level. Pyruvate is a metabolite at a key junction of cellular energy metabolism allowing to probe glycolysis and the TCA cycle. In contrast, pH, another key parameter for disease diagnosis⁴ lacks suitable hyperpolarized imaging agents which are non-degrading, easy-to-polarize and exhibit long-lived hyperpolarized ¹³C-labels. While [1,5-¹³C₂]zylonic acid has been introduced as a promising agent to image pH, it still comes with severe downsides such as its poor stability in aqueous solutions and the proximity of its pH sensitive chemical shifts to its decay products, making spectral analysis complex and error-prone. Another pH agent that has been introduced, hyperpolarized bicarbonate, suffers from short T₁ relaxation times and complicated polarization protocols⁵ while not being exclusively extra- or intracellular. For bicarbonate, the ratio of two signals is used for the calculation of pH, which requires careful peak quantification and permits only the detection of an average pH per voxel which is weighted unknown intra- and extracellular fractions. Given the success of [1-¹³C]pyruvate in the field of hyperpolarized ¹³C-MRI, we considered pyruvate-based and structural alike molecules (e.g. α-keto-acids) to be promising for pH imaging, but in the past suitable synthesis routes were missing. This changed by the work of Rios et al.⁶ which opened the way to access several pyruvic acid-derivatives synthetically. Given the variety of derivatives that can be synthesized, we chose the most promising one (Z-OMPD) based on its favourable energy state, high potential biocompatibility and achievable synthesis yield.

To highlight both the unique properties of Z-OMPD compared to other pH sensors and our motivation for its synthesis, we also extended the results chapter as suggested:

P2, L25-32: “Given the lack of non-degrading, easy-to-polarize, chemical shift-based and long-lived hyperpolarized ¹³C-labelled pH sensors⁴⁶, recently emerging synthesis routes and structures of pyruvic acid-derived molecules⁴⁷ were analyzed for molecules with favourable hyperpolarization properties and T₁ relaxation times⁴⁸. Here, out of this molecule class, previously introduced [1,5-¹³C₂]zylonic acid⁴¹ exhibits a pK_a in the physiological range and good hyperpolarization properties, but requires co-polarization and degrades. We identified Z-OMPD as the pyruvic acid dimer with the most favourable pH-sensing properties having the lowest energy level of all isomers of pyruvic acid dimers and high biocompatibility^{43,47,49}. The C₁- and C₅-positions were considered to exhibit the most promising pH sensitivity and relaxation properties.”

Similarities and differences to zylonic acid and implications for clinical translatability:

While both molecules (zylonic acid (ZA) and Z-OMPD) can be applied for in vivo pH imaging, there are essential differences between them. We understand that it is of great importance to point out the properties which distinguish the molecules:

It is true that ZA and Z-OMPD are molecules that are related by the fact that both can be derived from different dimerization processes of pyruvic acid⁶. Thus, they exhibit the same molecular formula (C₆H₆O₅) and molecular weight (157 g/mol), leading to the initially included statement of similarity in the manuscript. However, ZA with its ring-like structure and Z-OMPD with its

open chain geometry exhibit strong structural differences, which lead to very different physical and chemical properties apart from their T_1 relaxation times (Fig. 1):

- ZA is a monocarboxylic lactone acid, while OMPD is a dicarboxylic acid. This is important for biological processes like membrane transport as dicarboxylic acids are more abundant in biological processes (such as fumarate, malate, succinate) and might indicate superior biocompatibility.
- For ZA and some dimers of pyruvic acid, synthesis and natural degradation of pyruvate into ZA has been reported for several years while a synthesis route for OMPD has been published for the first time in 2020⁶.
- ZA is unstable in aqueous solutions and degrades to parapyruvate hydrate with a half-life of 2.5 h⁷, while Z-OMPD is stable in water (Supplementary Figure S4). This is due to an energetically more favorable conformation of Z-OMPD (Figure 3 in Rios et al., 2020).
- Stability is of critical importance for clinical translation, since it immensely facilitates sample storage, increases the robustness of the hyperpolarization protocol and simplifies pH determination. For ZA, degradation peaks which contaminate the single-voxel spectra can exemplarily be seen in Fig. 3d of the respective publication⁷. Regarding regulatory requirements, degradation products require a more complex quality assurance protocol and tremendously more efforts to evaluate all safety aspects prior to application in humans which requires substantially more effort and financial resources to achieve translation and applicability.
- Importantly, the difference in the molecular conformations of ZA and Z-OMPD leads to different pH sensitivities. This results in the possibility to perform accurate pH imaging without external reference using Z-OMPD, which is a major progress in the approach and facilitates clinical translation for several reasons:
 - While pH imaging has only been shown to be robust with ZA using urea as a co-injected reference compound, the negligible pH shift of the OMPD- C_1 peak enables pH imaging without co-injection of reference compounds. Thus, for ZA, the signal lifetime of co-injected urea (which typically exhibits a lower polarization level and T_1 relaxation time) limits the pH imaging approach, due to the absence of a pH reference as soon as the urea signal has decayed.
 - Co-hyperpolarizing two agents (urea + ZA) further complicates the sample preparation process (the two compounds should not mix and therefore have to be frozen on top of each other using liquid nitrogen). It has been shown that co-polarization and dissolution of two agents (urea- and pyruvate) can lead to the formation of side products⁸. These side products are unwanted, potentially corrupt image acquisition and analysis as they contaminate the acquired spectra (Fig. 1c in Qin et al., 2022) and in the worst case even are harmful (section 2.7 in Qin et al., 2022).
 - The combined measurement protocol of pH, perfusion and renal filtration as shown in Fig. 5,6 our submitted manuscript would not be feasible with ZA or any other reported hyperpolarized agent. First, as outlined, urea is needed as co-injected agent for the quantification of pH using ZA. Second, perfusion imaging using e.g. the less-shifting C_5 -peak of ZA for a sole injection of ZA is unfeasible, because both of ZA's ^{13}C -labeled peaks shift strongly with pH and are in close proximity to each other, making a spectrally selective imaging approach (as used for our method) impossible without depleting the magnetization of the second peak and suffering from signal contaminations in the actually targeted peak depending on pH.
 - If for our novel study on hydronephrotic kidneys (see Figure 7 and first comment of Reviewer #2) we would hypothetically use ZA instead of Z-OMPD, only the pH compartments but not tGFR and tRBF could be measured. In addition, the reliable

detection of the third and diagnostically most relevant pelvic pH compartment would be questionable as the diseased kidneys show a lower filtration rate, thereby requiring more time for agent accumulation within this compartment, which competes with the shorter T_1 relaxation times of both ZA and urea compared to Z-OMPD. Therefore, a multiparametric analysis of the hydronephrosis model as shown with Z-OMPD would not be feasible with ZA.

Because of the aforementioned reasons we agree with the reviewer, that we should make a clear separation between the molecules and included several of the following points in the revised version of our manuscript:

P13, L16-22: "In particular comparison to zymonic acid with its ring-like geometry, the open-chain structure of Z-OMPD results in several advantages for pH imaging, including longer T_1 relaxation time constants, an internal frequency reference (avoiding the co-injection of a reference compound) and high stability in aqueous solution. These molecular properties, together with a dedicated set of tailored MR pulse sequences enabled the simultaneous read-out of three imaging biomarkers (pH, perfusion and renal filtration) within a single imaging session after one injection of Z-OMPD within one minute which is a unique feature of Z-OMPD as a molecular imaging agent."

Novelty of Z-OMPD synthesis?

Rios et al. were the first to synthesize (non-labelled) Z-OMPD in 2020⁶ from methylpyruvate as a crude product. Our synthesis approach described in Fig. 1a of the submitted manuscript is based on this work with the difference that we validated the entire synthesis route regarding intermediate and final product using ($[1-^{13}\text{C}]$)ethylpyruvate instead of methylpyruvate as a precursor which reduces synthesis cost for ^{13}C -labelled Z-OMPD by a factor of 3-5, thereby increasing accessibility for future works. The reaction processes are similar for both precursors, so we do not conclude that the synthesis is particularly novel compared to the referenced work. However, we developed a protocol for preparative HPLC-based purification using a C18 reverse-phase column and a linear gradient of acetonitrile with water, resulting in a very high purity ($\geq 96\%$) which is unmatched by Rios et al., (Rios et al., Supporting Figure S16 vs. our manuscript, Supplementary Figure S1). In conclusion, we provide an optimized version of an existing synthesis approach with respect to synthesis cost for the ^{13}C variant and chemical purity of the product (compare Rios et al., ACS Omega, 2020, Supporting Information VI).

Introduction

-paragraph 1, "In clinical routine, kidney function is probed by perfusion imaging using arterial spin labelling"? This is a research tool but not in standard clinical routine. Kidney function is routinely probed by iodinated contrast multi-phase CT and MAG3 nuclear scintigraphy using MAG3 as imaging agent, please rephrase.

The respective paragraph in the introduction has been corrected and clarified (P1, L31-35): "In clinical routine, kidney function is probed by iodinated contrast multi-phase CT^{8,9} or scintigraphy using MAG3¹⁰ as imaging agent. Recently introduced methods, not translated to clinical routine yet, involve the injection of Gd-based contrast agents for DCE-MRI¹¹, diffusion weighting¹² or arterial spin labelling^{13,14}. However, scintigraphy using $^{99\text{m}}\text{Tc}$ -MAG3 involves injection of ionizing radiation and [...]"

-paragraph 1, "MRI- and CT-based contrast agents can induce nephropathies,"gadolinium MR contrast doesn't really induce a nephropathy but instead there is a risk of "nephrogenic systemic fibrosis" which involves fibrosis of skin, eyes, joints and internal organs for patients who have impaired kidney function. Please rephrase.

This paragraph has also been corrected as follows (P1, L36-40): “Further, CT-based contrast agents can induce nephropathies¹⁵, which poses a major risk for single or repeated assessment of already fragile kidney conditions while MRI-based contrast agents can accumulate in organs¹⁶, bear a risk of inducing nephrogenic systemic fibrosis in patients with impaired kidney function¹⁷ and are suspected to accelerate metastasis¹⁸.”

Apart from the known risk for nephrogenic systemic fibrosis, we also want to point out to a recent study where it has been shown, that Gd potentially accelerates metastases⁹.

-The first few sections in Results are quite choppy and partially read like methods sections instead, which they shouldn't. They should smoothly describe the premise of the experiments performed and then report sequentially the observations that can be made based on the figures and tables. Please improve this. For example, for paragraph 1, sentence 1 should mention what initiated the idea to synthesize Z-OMPD as a hyperpolarized agent for pH imaging then describe the details of the synthesis yields and isotopic enrichment levels.

The first four sections of the results chapter were carefully checked for readability and improved accordingly (P2, L24-25; P2, L37; P3, L16-17; P4, L18-19; P5, L15-17). Particularly the first section regarding the synthesis of ¹³C-labelled Z-OMPD was extended and several points that drove the project initiation are now highlighted (P2, L24-36):

“Synthesis of Z-OMPD. Given the lack of non-degrading, easy-to-polarize, chemical shift-based and long-lived hyperpolarized ¹³C-labelled pH sensors⁴⁶, recently emerging synthesis routes and structures of pyruvic acid-derived molecules⁴⁷ were analyzed for molecules with favourable hyperpolarization properties and T₁ relaxation times⁴⁸. Here, out of this molecule class, previously introduced [1,5-¹³C₂]zymonic acid⁴¹ exhibits a pK_a in the physiological range and good hyperpolarization properties, but requires co-polarization and degrades. We identified Z-OMPD as the pyruvic acid dimer with the most favourable pH-sensing properties having the lowest energy level of OMPD isomers and high biocompatibility^{43,47,49}. The C₁- and C₅-positions were considered to exhibit the most promising pH sensitivity and relaxation properties. The recently published synthesis route⁴⁷ was modified by using [1-¹³C]ethyl pyruvate as a precursor to synthesize [1,5-¹³C₂]Z-OMPD (Fig. 1a, for details see methods). Product purity improved by development of a HPLC-based purification protocol with a final yield of ~18% (weight fraction) and a purity of ≥96% (Fig. S1). The structure and mass of unlabelled Z-OMPD and ¹³C-labelled [1,5-¹³C₂]Z-OMPD were confirmed by ¹H-NMR spectroscopy and mass spectrometry (Fig. S2) respectively.”

-Results, pH imaging in healthy rat kidneys section, When there are literature values for the pH of compartments mentioned in results, they should be explicitly mentioned, i.e. cortex was previously pH = xxx, renal pelvis pH = yy instead of simply that your measurements “agree with previous studies”

The literature values for all three compartments for both references are now explicitly stated in the manuscript (P5, L1-2): “[...] agreeing with previous studies^{41,50} where pH_{Cortex} = 7.30 – 7.40, pH_{Medulla} = 6.94 – 7.00 and pH_{Pelvis} = 6.3 – 6.55 was reported.”

-Figure 1, The legend and caption for figure 1d are a little confusing, please mention what ZAd,1 and ZAd,5 are compared to OMPD1 and OMPD5 in either the figure caption which is presumably zymonic acid and Z-OMPD. It would be nice to introduce zymonic acid's structure and describe the results of this compared to Z-ompd in paragraph 2 of results where the invitro pH sensitivity is described. This is also related to the chopiness of the Results section.

The legend in Fig 1d has been clarified by explicitly referencing the abbreviations ZAd_{d,1/5} and OMPD_{1/5}: (P3, L12-13): “Comparison of T₁ relaxation time curves of ¹³C-labelled hyperpolarized in vivo pH sensors shows ¹³C-labels of undeuterated Z-OMPD (OMPD₁; OMPD₅) to exhibit superior hyperpolarized signal lifetime compared to deuterated zymonic acid (ZAd_{d,1}; ZAd_{d,5}).”

Since the structure of zymonic acid has already been reported, can be easily accessed in literature⁷ and the molecule is only of minor interest for this study, we prefer to not explicitly include the structure as a figure in the manuscript. However, we support the reviewers point of a direct comparison of important parameters and added explicit values for T_1 relaxation times (P3, L19-21): “[...] longer (C_1 : 138 s, C_5 : 119 s) than those of alternative hyperpolarized in vivo pH sensors like [^{13}C]bicarbonate⁴⁰ or [1,5- $^{13}\text{C}_2$,3,6,6,6- D_4]zymonic acid (ZA_d ; C_1 : 47s, C_5 : 75s) (Fig. 1d).”

-Figure 2 (labeled as a second Figure 1), please mention the buffer and concentration in the legend for Figure 2c instead of just stating “buffer”. For the others, the concentrations and items are more explicitly mentioned. Also, is this taken to be at a different concentration in Z-OMPD than 10 mM or 100 mM which are also listed?

The legend of figure 2c has been adjusted to clarify that a citrate-phosphate buffer prepared from stock solutions of either 200 mM Na_2HPO_4 or 100 mM Citric Acid was used where the exact composition depends on the desired pH of the final buffer solution (https://en.wikipedia.org/wiki/McIlvaine_buffer). In addition, the citrate-phosphate buffer system is now also termed in the figure caption of figure 2 (P4, L2) and in the text (P4, L11). In fact, the citrate-phosphate buffer- and blood data points originate from the phantom imaging acquisitions in Fig. 2a and b which are 6 mM and 11 mM respectively. This is now also clarified in the figure caption (P4, L1-8):

“Figure 2 | *In vitro* pH imaging of Z-OMPD in buffer and blood phantoms. (a) pH map generated from a CSI acquisition of 6 mM Z OMPD and 10 mM [^{13}C]urea in six citrate-phosphate buffer phantoms of different pH values, overlaid with a corresponding T_2 -weighted image. Black numbers indicate reference phantom pH values measured using a conventional pH electrode. **(b)** pH image of three human blood phantoms of different pH values after injection of 11 mM hyperpolarized Z-OMPD and 19 mM [^{13}C]urea, overlaid with a T_1 -weighted image. White numbers display phantom reference pH values. The white center phantom (a) and top phantom (b) contain only thermal [^{13}C]urea for B_1 calibration. **(c)** pH values measured under various citrate-phosphate buffer conditions (phantom conditions from imaging in a), blood (phantom conditions from imaging in b), pH sensor-, salt- or protein concentrations using Z-OMPD agree well with measurements using a conventional pH electrode.”

Further details regarding the preparation of all phantoms are also listed in the respective method section “Buffer and human blood phantoms” (P14, L54 – P15, L9).

-Figure 3, This figure could use the T2w image showing the kidneys for panel d and e, please display it. Particularly since it is hard to see where the cortex, medulla and pelvis are without this.

In combination with comment #1 by Reviewer #3, we added an inset for figure 3c which shows the anatomical T_{2w} image for the panels in d and e, together with the respective voxel location of the spectrum in Fig. 3c. The figure caption has also been modified to point out this connection (P5, L5-6): “[...] Exact voxel position is indicated in the inset with a yellow square, the T_{2w} image corresponds to the background in d and e. [...]”

-It is a bit weird structurally to have a section for pH of healthy rat kidneys then another for pH of tumors and then go back to kidneys and have a section for pH and perfusion of healthy rat kidneys, why wouldn't the pH and the perfusion of kidneys sections be back to back?

We do comprehend the reviewers point of view to place the kidney-related sections back to back because they share the same imaged organ. The idea behind our structure of this manuscript is to present the results both in a methodological and in an order of increasing complexity. Regarding methodological order, this means to start with synthesis, characterization, validation of a novel pH sensor in vitro and in vivo (healthy kidney imaging) and then proceed to the application of pH imaging in tumors. Then we present combined pH

and perfusion imaging adding another quantitative readout. With the new pathological hydronephrosis model (Fig. 7) this order adds a combined perfusion and pH imaging protocol of a pathological model as the last example of highest complexity. We intend to clarify this by the introductory sentences of the respective sections:

P4, L25: “Validation of *in vivo* pH imaging was done in healthy rat kidneys which possess physiological tissue acidification due to the renal filtration process.”

P6, L14-15: “To apply the validated *in vivo* pH imaging in an oncological setting, [...]”

Given that we added another chapter where we applied the simultaneous perfusion, filtration and pH imaging protocol in a model of hydronephrosis (section “Multiparametric imaging of hydronephrosis in kidneys caused by pheochromocytoma in MENX rats.”), we would like to point out that the same order of first validating and then applying the new, more complex imaging protocol is also employed for the latter chapters. We highlight this by respective introductory sentences at the beginnings of the key chapters:

P7, L13-14: “Beyond sole pH imaging, Z-OMPD can be used for combined imaging of pH and perfusion.”

P9, L37-40: “To demonstrate the value of measuring up to five parameters (RBF, GFR, $\text{pH}_{\text{Cortex}}$, $\text{pH}_{\text{Medulla}}$, $\text{pH}_{\text{Pelvis}}$) within one minute and a single injection, simultaneous imaging of perfusion, filtration and pH by hyperpolarized ^{13}C -MRI of Z-OMPD was employed to probe the kidney function state in MENX rats^{58,59} [...]”

-Figure 4, Is it not clear from the images in panels a, c that the ROI is in a muscle compartment, is it possible that it is instead an ROI in the peritoneal cavity or intestines or other compartment? It seems a little odd if it is muscle firstly that there is much contrast in muscle and also that it is 0.2 units lower in pH than the muscle for the other animals? The observation could be explained by the ROI not being muscle but something else instead as there is probably a lot of free probe outside the muscle in the peritoneal cavity.

As stated in the figure caption (P6, L3-4), the white square in Fig. 4a is not a muscle ROI, but rather an arbitrarily placed indicator of the native voxel size, to illustrate the true spatial resolution of the imaging acquisition. However, we understand that the previous placement of this voxel can give rise to confusions and relocated the white square apart from any area containing tissue to avoid any ambiguity. For further clarity, the muscle ROI for the animal in Fig. 4 is shown below:

In general, muscle ROIs were always drawn on spine muscle regions of sufficient SNR (>3) to obtain reliable pH values. Regarding the differences of up to 0.2 pH units between some animals, it has been shown, that prolonged isoflurane-anaesthesia, as it is required for these imaging studies, can lead to severe respiratory acidosis (Reference 51-55 in the submitted manuscript) in mice, which is within the range that we observe for the drops in muscle pH. Also

since we do not observe a systemic deviation from the physiologic pH range across all animals but see inter-subject variation, anesthesia appears to be a likely source of this finding.

-Figure 4, Are there differences in the time course for observing this second pH compartment for the lymphoma tumors or do both compartments come up at similar timings after injecting Z-OMPD? This might be the case, it would be nice to understand if it is or not.

The question about the influence of the injection and acquisition for compartmental pH imaging is definitely a very important one. We performed pilot experiments in some of the animals where we injected the same animal twice with hyperpolarized Z-OMPD, once with 9s between end of injection and start of image acquisition and once with a 15s delay. We have added a figure to the supplement to address this point in more detail in addition to a clarifying sentence in the manuscript (P6, L18-20): "Here, mean pH and compartment detection are potentially influenced by tracer extravasation into the extracellular space and signal decay (Fig. S7).

Fig.S7

Supplementary Figure 7 | Temporal dependence of compartment detection and mean pH.

(a) The observation of two pH compartments within tumor tissue can be explained by the pH sensor being located either in vascular or extracellular space. For voxels, containing predominantly vessel signal, only one compartment can be observed (left spectrum), while voxels, covering also acidified extracellular space with sufficient tracer accumulation, a second peak, representing this tissue compartment can be observed (right spectrum). As the mean pH is weighted by the signal intensities from both observed compartments, delayed accumulation in the extracellular space compared to the fast perfused vascular space results in acquisition delay dependent mean pH contrast **(b, c)**. The physiological compartment **(d, e)** is present at both acquisition timepoints (9 s and 15 s post end of injection) due to the rapid perfusion whereas Z-OMPD is detectable on more tumor areas with an acidified extracellular space for later time points **(f, g)**. Here, increasing extravasation of Z-OMPD to the extracellular space with increasing acquisition delay competes with hyperpolarized signal decay for optimized imaging time points. Nevertheless, pH values for each compartment **(d-g)** agree with each other for both acquisitions, making pH compartment images quantitatively more robust with respect to timing variations compared to mean pH maps.

The first compartment is rather vascular and quickly perfused and appears always in almost the entire tumor. The second compartment seems to be better observable after 15s. The major limitation here is that the second, rather extracellular compartment (scheme below) accumulates Z-OMPD slower and in smaller quantities, while the hyperpolarized signal decays with T_1 . In order to reliably fit the second compartment peaks, sufficient SNR is necessary where the uptake and signal decay are two competing processes. Nevertheless, while the signal intensities of the two compartments, and therefore the compartmental weighting for the mean pH is indeed sensitive to these timing aspects, the absolute pH values for each compartment depend only on the chemical shift and not the peak intensity (as long as detectable). This explains, why the tumor mean pH in Fig. 4f shows more variation compared to the individual compartments and why compartmental pH imaging is more robust compared methods which capture only the mean pH such as hyperpolarized [^{13}C]bicarbonate or CEST.

- The time courses in figure 5d and 6 a-d seem a little short for measuring clearance of probe based on the prior literature using CT and MAG3 scintigraphy as this should be measured over minutes instead of only 30 s. Granted that the polarization persists for at $t_{1/2}$ of 30 s in vivo based on Figure 1e, but there is also not a good validation of the finding.

While we agree with the reviewer that the used time interval of 30 s provides a lower limit for the time interval to image renal filtration, we want to point out, that major renography analysis methods, such as the Patlak-Rutland model¹⁰ and the slope method¹¹ rely on the early uptake phase within the kidneys. Even for humans as larger subjects, only the first 2-3 minutes represent the relevant uptake phase^{11,12} and international consensus also suggests to not include any data later than 2 min 30s post injection into the quantification¹³. Further, comparison of different analysis intervals for GFR analysis has been shown to be of a smaller effect^{14,15} compared to the choice of analysis method¹⁶. Here, the Baumann-Rudin model, which is also used in our study even showed the best performance for determining GFR values while only requiring the shortest time interval for data fitting. Moreover, the major purpose of acquiring longer time intervals is to determine residence times of the tracer in the kidney ($T_{-1/2}$). For this parameter, no evidence¹⁷ has been found that it is of diagnostic value nor any standardization for its calculation¹⁸. In addition, these considerations are generally made for human subjects. Relation of normal GFR for a human (~ 120 ml/min) to its total blood volume (~ 5 l) compared to a rat (GFR ~ 5 ml/min, blood volume ~ 15 ml) indicates, that rat kidneys have a 14-fold higher filtration performance compared to humans, which explains why we see the filtration process already after a shorter time period compared to humans which is independently found by imaging (Fig. 5d) and spectroscopy (Fig. 6k). Nevertheless, we agree

with the reviewer that for studies in humans, the imaging time to assess renal filtration should be prolonged for reliable GFR quantification.

Regarding the validation of Z-OMPD's *in vivo* T_1 time constants, we would like to draw attention to the method section (P15, L33-36) where we describe that T_1 relaxation time constants were explicitly measured from the data that was used for slice spectroscopy-based calculation of GFRs, in line with measurements of other studies to determine T_1 of hyperpolarized agents *in vivo*^{4,7}.

-Results for fig 5,6, It would be nicer to mention the measured GFR literature numbers at least in the discussion and describe how similar or different they are for your experiments instead of generally mentioning they agree but not listing numbers.

For direct comparison, literature values for both tGFR and tRBF have been inserted into the results section where values from this study are reported:

P9, L21-23: "Nevertheless, tGFRs agree reasonably well with previous studies³⁵, where methods using hyperpolarized [¹³C]urea (tGFR = 5.1 ± 0.9 ml/min), DCE-MRI (tGFR = 3.5 ± 1.5 ml/min and Inulin (GFR = 5.1 ± 0.9 ml/min) are reported."

P9, L26-27: "Analogous to tGFR, tRBF values agree with previous studies³⁵ using DCE-MRI (tRBF = 9.5 ± 3.1 ml/min /ml tissue) with better agreement for Z-OMPD compared to urea."

Reviewer #2 (Remarks to the Author):

The manuscript by Grashi et al. "Simultaneous Magnetic Resonance Imaging of pH, Perfusion and Renal Filtration using Hyperpolarized ^{13}C -labelled Z-OMPD" describes a novel pH imaging biomarker based on hyperpolarized ^{13}C MRS. The biomarker is novel, however the use case of renal cancer, is less so and the additional GFR and perfusion estimation is not utilized in any significant way. pH is important, GFR is important, and perfusion is important, but the paper fails in describing why the simultaneous acquisition is important?

We agree with the reviewer that the previous version of the manuscript did not comprehensively enough describe the importance of the simultaneous acquisition of pH, GFR and renal blood flow.

To assess the diagnostic value of the combined set of these parameters, we applied the simultaneous acquisition protocol of RBF, GFR and pH using Z-OMPD in a preclinical model of MENX rats¹⁹ which develop p27-mutation-induced pheochromocytoma. Due to the tumor-induced increased release of catecholamines and calcium, high blood pressure and mineralization damage the kidneys which leads to partial ureteral obstruction and manifestation of clinically authentic hydronephrosis. All new study results are included the results chapter as a separate section, including Fig.7 and Fig. S15) of our resubmitted version of the manuscript:

“Figure 7 | Comparison of kidney parameters in hydronephrosis measured by hyperpolarized ^{13}C -MRI using Z-OMPD and clinical standard techniques. (a) 3D dynamic perfusion imaging of hydronephrotic kidneys shows poor filtration of Z-OMPD towards the renal pelvis (grey arrows) despite strong cortical perfusion (white arrows). This is quantitatively reflected by normal $t\text{RBF}$ values (b) but strongly reduced $t\text{GFR}$ values (c). Subsequent pH imaging shows homogeneous and physiological pH compartments for the cortex (d, e) and the medulla (f, g) while the pelvis exhibits a strong, pathologic acidification (h, i). Standard MRI methods for assessment of renal disease status show a moderate increase (j) in hydronephrotic kidney ADC (diffusion-weighted MRI) compared to healthy controls and a substantial volume (k) increase of hydronephrotic kidneys (T_2 -weighted MRI). (e) Assessment of renal biomarkers in blood serum reveals mildly increased creatinine- (l) and strongly elevated urea level (m) while SDMA values are not clinically evident (n). Reference ranges (dashed lines) were obtained from animal suppliers for creatinine and urea⁵⁹ or literature for SDMA⁶⁰⁻⁶². (o) Comparison of all parameters showing significant evidence for hydronephrosis using Cohen’s d indicates pelvic pH and $t\text{GFR}$ measured by hyperpolarized imaging Z-OMPD to be most sensitive to this kidney disease. Scale bars and white squares in a and d indicate 10 mm and native acquisition resolution respectively.”

Our perfusion and filtration imaging data shows no alterations in renal blood flow (RBF) but reveals a reduced GFR. pH imaging shows a reduced pH in the pelvic kidney compartment compared to control rats, while the medullary and cortical pH compartment are within physiological ranges. This results in two pathologically altered and three unaltered, physiological parameters (Fig. 7). The joint alteration of GFR and pelvic pH appears reasonable since the filtration process is the cause for the ultimately well-defined pH compartments towards the pelvis due to an increased waste product/acid concentration further along the filtration process. A disturbed filtration process is therefore suitable to sensitively affect the pH compartments. Furthermore, the evaluation by histopathology and mass spectrometry imaging agrees extraordinarily well with our findings (Fig. S15).

Here, two pathologically altered kidney parameters yield a stronger statement about the compromised health of the kidneys compared to having just one single parameter. Two independent imaging biomarkers pointing towards diseased kidneys are generally more credible than just one single image-derived parameter.

However, there might also be diseases, where tissue acidification is caused by metabolic changes rather than by disturbed GFR. In these cases, the combined parameterset of pH, RBF might return different combinations of pathologically and physiological parameters, yielding a more comprehensive image of the overall kidney health which might allow kidney disease classification based on its parameter “fingerprint”.

Beyond nephrology, pathologies involving cancer, metabolic diseases and altered perfusion often exhibit opposite changes in pH and perfusion in disease progression and therapeutic response^{8,20,21}, thus getting readouts related to both parameters is a goal of many currently evolving imaging techniques, such as combining hyperpolarized pyruvate and urea⁸ or combining different imaging modalities on separate devices to get pH and perfusion readout²². However, those protocols either requiring co-polarizations or 30-fold longer acquisition times (> 30 min) compared to our protocol.

Many different perfusions and GFR estimators exist, yet few enter clinical use and the fact that the data is acquired at the same time, has little real impact in the clinical setting. eGFR is still today the most used GFR measure. Renal MRI perfusion is not a very used clinical tool, although it has great potential.

We acknowledge the reviewer’s point of view, that it is challenging to translate perfusion and GFR estimators into the clinic, particularly if they rely on imaging techniques. However, the accuracy, reliability and value of eGFR as predictor for classifying kidney patients and therapy

selection is heavily debated²³⁻²⁵. We therefore want to point out the complementary value of our Z-OMPD-based MRI biomarkers to eGFR as a clinical standard of care:

eGFR cannot discriminate between single kidneys, which limits the sensitivity and specificity of the biomarker. Thus, imaging-based techniques are needed for individual kidney function assessment. However, many MR-based approaches use potentially harmful gadolinium-based contrast agents, the CT-based approaches involve iodinated contrast agents plus ionizing radiation and the standard of care, ^{99m}Tc-MAG3, uses radioactivity while only acquiring 2D projections. Our approach offers a fast and radiation-free alternative without the involvement of gadolinium or iodine-based contrast agents while delivering 3D perfusion and 2D pH information at the same time.

eGFR depends on parameters that fluctuate strongly between individuals, such as muscle mass²⁶, diet²⁷ and certain drugs²⁸. Additionally, eGFR has been shown to be an insufficient measure for acute kidney injury²⁹.

As described above, our newly included study on hydronephrosis shows, that the assessment of pH, perfusion and filtration is useful for assessment of kidney health status, while the creatinine level was insufficiently sensitive to the pathological hydronephrotic state (Fig. 7I). Notably, in our conducted study (see response to previous comment), all serum kidney function parameters (creatinine, SDMA, urea), ADC and kidney dilatation are less sensitive (lower Cohen's d) compared to tGFR and pelvic pH in detecting the present kidney damage, rendering the multiparametric imaging protocol using Z-OMPD superior compared to clinical routine methods, including eGFR. In particular, for kidney patients where eGFR estimates are "on the brink" of requiring treatment, our imaging technique might be valuable to stratify those patients and preventing overtreatment or delayed treatment onset.

We have extended the discussion regarding the abovementioned points.

In summary the most noteworthy results are the novel pH sensor, it has as described in this manuscript little impact, as the combination adds little value, and the data here does not show any benefit over any other pH measurements, in fact the compartment pH data (figure 4) suggest low specificity. The authors claim significant downfalls of CEST and other techniques and thus it would be beneficial to demonstrate this or demonstrate that the unique simultaneous nature of the RBF/GFR/pH measurement has a benefit over existing methods or yields a completely new information not before obtainable. I recommend a major revision with a requirement to add additional data on the pH biomarker (comparison to alternative techniques) or the combination of pH/GFR/perfusion.

First, we appreciate the reviewer's overall positive feedback. Regarding the claimed low specificity, we assume that the reviewer refers to the pH compartment maps showing some tumor areas exhibiting both (physiological and acidified) compartments. This is not a low specificity but in fact related to the microscopic heterogeneity of the tumor tissue bearing small vessels and acidified extracellular space. A schematic drawing of the tissue model explaining our data is display below:

The observation of up to two pH compartments within tumor tissue can be explained by the pH sensor being located either in vascular or extracellular space. For voxels, containing predominantly vessel signal, only one compartment can be observed (left spectrum), while voxels, covering also acidified extracellular space with sufficient tracer accumulation, a second peak, representing this tissue compartment can be observed (right spectrum). We also added a figure to the supplement (in relation to a comment from Reviewer #1) which assesses the compartment aspect in more detail (Fig. S7).

We do not share the opinion of the reviewer that the most noteworthy results are the novel pH sensor and want to point out that the combined, fast (sub-minute) assessment of several imaging parameters (pH, GFR, RBF) and the given and molecular properties and opportunities (internal reference, widely spaced chemical shifts for clean RF excitation, long T_1) require a careful tailoring of MR acquisition and analysis protocols which demonstrates technical MRI imaging advances far beyond so far introduced sole pH imaging protocols.

To combinedly assess those parameters, previous studies used complicated, time-consuming protocols²² where they measured those three biomarkers consecutively using MRI to measure pH and perfusion, followed by multispectral optoacoustic tomography to assess filtration or lengthy CEST protocols³⁰. While these approaches are interesting and novel, they require a lot of time and would present a high organizational and logistical burden if translated to the clinic. In this context, saving time is of high clinical interest, due to the tremendous requirement of personnel, difficulties to recruit patients (the longer the patients are required to remain in the clinics, the more difficult they are to recruit) and the general pressure of saving health care resources.

In addition, in our new study ("Multiparametric imaging of hydronephrosis in kidneys caused by pheochromocytoma in MENX rats."), application of our multiparametric imaging protocol results in five diagnostically relevant parameters ($tRBF$, $tGFR$, pH_{Cortex} , $pH_{Medulla}$, pH_{Pelvis}) of which two show a pathologic alteration while the remaining three parameters are within physiological ranges. Further studies of other kidney diseases using this protocol would be of high interest, as those might reveal, whether certain patterns of pathological alterations in those parameters provide a unique fingerprint of the respective kidney disease, thereby facilitating diagnosis. In light of the increasing requirement to combine multiple imaging modalities and derive more than one parameter at the same time to maximize diagnostic value³¹ while saving time and in consequence money, our method fulfils these requirements of providing five diagnostic parameters while requiring only one minute acquisition time, a feature which, to our knowledge, no other imaging modality can achieve at the moment. Additionally, CEST-based pH measurements and hyperpolarized bicarbonate are inherently measuring compartment-weighted mean pH values when the tissue pH compartment heterogeneity is below the imaging resolution of the respective technique. As the pelvic area contains major blood vessels, the poorly accumulating but for hydronephrosis studies critical pelvic pH compartment would rather contribute minorly to the mean pH, rendering the observation of this pelvic acidification difficult with CEST, bicarbonate or any other non-compartment-sensitive pH imaging technique, while zymonic acid might suffer from peak overlap of C_1 - and C_5 -resonance for these strongly acidified pH values. We extended the discussion regarding some of the mentioned points (P12, L47 – P13, L2):

"Nevertheless, to our knowledge, no other non-invasive imaging technique allows such a fast and comprehensive assessment of kidney function parameters as perfusion techniques are unable to assess pH and CEST-based approaches to combine pH and perfusion imaging fail to capture the individual pH compartments and rather return a mean pH. For such methods the critical pelvic pH compartment of our study would contribute with minor weight to an observed mean pH, rendering the observed acidification difficult to assess."

Reviewer #3 (Remarks to the Author):

Grashei et al. present the manuscript entitled “Simultaneous Magnetic Resonance Imaging of pH, Perfusion and Renal Filtration using Hyperpolarized ¹³C-labeled Z-OMPD. Overall, this is an exciting, rigorous, and well written manuscript presenting data which will be of significant interest to the community of researchers engaged in the development of pH imaging methods and hyperpolarized probes. The conclusions are clearly supported by the data presented. No major flaws in data analysis or presentation were noticed.

We thank the reviewer for the positive feedback and appreciate that he shares our point of view.

Some minor edits are suggested:

1. For figure 3c, where is this single voxel location on the rat kidney?

Figure 3c is now complemented by an inset of the respective anatomical MRI image and a yellow square indicating the single voxel location.

2. The discussion is a “wall of text.” Would recommend to reformat a bit to break down into three paragraphs or so.

The discussion is now reformatted into seven paragraphs which closely align with the order of the presented results.

3. Page 1 - L14. T1 at what magnetic field strength?

The corresponding field strength is now stated (P1, L14): “of two minutes at 1 T”

4. Check the manuscript throughout to use consistent nomenclature throughout. In some cases [¹³C] is used, in other cases the brackets are not present.

The nomenclature has been checked and terms directly referring to a ¹³C-label within a molecule are stated with brackets now (changes mainly apply to [¹³C]urea).

5. Page 2 L4. Add following reference with 31. It was the first patient study. Liu, X. et al. Development of specialized magnetic resonance acquisition techniques for human hyperpolarized [¹³C,¹⁵N₂]urea + [1-¹³C]pyruvate simultaneous perfusion and metabolic imaging. Magn. Reson. Med. 88, 1039–1054 (2022).

The now respective reference (36) has been added.

6. The abbreviation definitions, like Z-OMP, IHC, RBF or others are also in Methods, they may need to move to the first time they appear.

Abbreviations are now defined where they appear the first time in the manuscript.

7. Replace heading of “pH imaging in tumors in mice” to something more specific to detail what is entailed in the section.

The section title was changed to (P6, L14): “Imaging of tumor acidification in subcutaneous EL4 lymphoma in mice.”

8. Page 8 L2. Is “quite squares” – I assume this means white squares?

Corrected.

9. Page 9 L43. “no sign of toxicity” up to what the maximum concentration of probe is.

15 mM as maximum tested concentration is now stated (P11, L44-45): “[...] up to 15 mM”.

References:

- 1 Ostergaard Mariager, C. *et al.* Can Hyperpolarized ^{13}C -Urea be Used to Assess Glomerular Filtration Rate? A Retrospective Study. *Tomography* **3**, 146-152, doi:10.18383/j.tom.2017.00010 (2017).
- 2 Liu, X. *et al.* Development of specialized magnetic resonance acquisition techniques for human hyperpolarized [^{13}C , $^{15}\text{N}_2$]urea + [$1\text{-}^{13}\text{C}$]pyruvate simultaneous perfusion and metabolic imaging. *Magn Reson Med* **88**, 1039-1054, doi:10.1002/mrm.29266 (2022).
- 3 Keshari, K. R. & Wilson, D. M. Chemistry and biochemistry of ^{13}C hyperpolarized magnetic resonance using dynamic nuclear polarization. *Chem Soc Rev* **43**, 1627-1659, doi:10.1039/c3cs60124b (2014).
- 4 Koltai, T. The Ph paradigm in cancer. *Eur J Clin Nutr* **74**, 14-19, doi:10.1038/s41430-020-0684-6 (2020).
- 5 Korenchan, D. E. *et al.* Dynamic nuclear polarization of biocompatible ^{13}C -enriched carbonates for in vivo pH imaging. *Chem Commun (Camb)* **52**, 3030-3033, doi:10.1039/c5cc09724j (2016).
- 6 Rios, A. C., Bera, P. P., Moreno, J. A. & Cooper, G. The Pyruvate Aldol Condensation Product. A Metabolite that Escaped Synthetic Preparation for Over A Century. *ACS Omega* **5**, 15063-15068, doi:10.1021/acsomega.0c00877 (2020).
- 7 Düwel, S. *et al.* Imaging of pH *in vivo* using hyperpolarized ^{13}C -labelled zymonic acid. *Nat Commun* **8**, 15126, doi:10.1038/ncomms15126 (2017).
- 8 Qin, H. *et al.* Clinical translation of hyperpolarized ^{13}C pyruvate and urea MRI for simultaneous metabolic and perfusion imaging. *Magn Reson Med* **87**, 138-149, doi:10.1002/mrm.28965 (2022).
- 9 Rahrmann, E. P. *et al.* The NALCN channel regulates metastasis and nonmalignant cell dissemination. *Nat Genet* **54**, 1827-1838, doi:10.1038/s41588-022-01182-0 (2022).
- 10 Rutland, M. D. A comprehensive analysis of renal DTPA studies. I. Theory and normal values. *Nuclear Medicine Communications* **6**, 11-20 (1985).
- 11 Moonen, M., Jacobsson, G., Granerus, G., Friberg, P. & Volkmann, R. Determination of split renal function from gamma camera renography: a study of three methods. *Nuclear Medicine Communications* **15**, 704-711 (1994).
- 12 Jouret, F. *et al.* Single photon emission-computed tomography (SPECT) for functional investigation of the proximal tubule in conscious mice. *Am J Physiol Renal Physiol* **298**, F454-460, doi:10.1152/ajprenal.00413.2009 (2010).
- 13 Prigent, A. *et al.* Consensus Report on Quality Control of Quantitative Measurements of Renal Function Obtained From the Renogram: International Consensus Committee From the Scientific Committee of Radionuclides in Nephrology *Seminars in Nuclear Medicine* **29**, 146-159 (1999).
- 14 Hackstein, N., Kooijman, H., Tomaselli, S. & Rau, W. S. Glomerular filtration rate measured using the Patlak plot technique and contrast-enhanced dynamic MRI with different amounts of gadolinium-DTPA. *J Magn Reson Imaging* **22**, 406-414, doi:10.1002/jmri.20401 (2005).
- 15 Piepsz, A., Tondeur, M. & Ham, H. Relative $^{99\text{m}}\text{Tc}$ -MAG3 Renal Uptake: Reproducibility and Accuracy. *The Journal of Nuclear Medicine* **40**, 972-976 (1999).
- 16 Bokacheva, L., Rusinek, H., Zhang, J. L., Chen, Q. & Lee, V. S. Estimates of glomerular filtration rate from MR renography and tracer kinetic models. *J Magn Reson Imaging* **29**, 371-382, doi:10.1002/jmri.21642 (2009).
- 17 Banks, K. P., Farrell, M. B. & Peacock, J. G. Diuretic Renal Scintigraphy Protocol Considerations. *J Nucl Med Technol*, doi:10.2967/jnmt.121.263654 (2022).
- 18 Taylor, A. T. *et al.* SNMMI Procedure Standard/EANM Practice Guideline for Diuretic Renal Scintigraphy in Adults With Suspected Upper Urinary Tract Obstruction 1.0. *Semin Nucl Med* **48**, 377-390, doi:10.1053/j.semnuclmed.2018.02.010 (2018).
- 19 Wiedemann, T. *et al.* Morphology, Biochemistry, and Pathophysiology of MENX-Related Pheochromocytoma Recapitulate the Clinical Features. *Endocrinology* **157**, 3157-3166, doi:10.1210/en.2016-1108 (2016).

- 20 Bok, R. *et al.* The Role of Lactate Metabolism in Prostate Cancer Progression and Metastases Revealed by Dual-Agent Hyperpolarized ¹³C MRSI. *Cancers (Basel)* **11**, doi:10.3390/cancers11020257 (2019).
- 21 Dunnwald, L. K. *et al.* Tumor metabolism and blood flow changes by positron emission tomography: relation to survival in patients treated with neoadjuvant chemotherapy for locally advanced breast cancer. *J Clin Oncol* **26**, 4449-4457, doi:10.1200/JCO.2007.15.4385 (2008).
- 22 Minhas, A. S. *et al.* Measuring Kidney Perfusion, pH, and Renal Clearance Consecutively Using MRI and Multispectral Optoacoustic Tomography. *Mol Imaging Biol* **22**, 494-503, doi:10.1007/s11307-019-01429-z (2020).
- 23 Badrick, T. & Turner, P. The Uncertainty of the eGFR. *Indian J Clin Biochem* **28**, 242-247, doi:10.1007/s12291-012-0280-1 (2013).
- 24 Porrini, E. *et al.* Estimated GFR: time for a critical appraisal. *Nature Reviews Nephrology* **15**, 177-190 (2019).
- 25 Levey, A. S., Coresh, J., Tighiouart, H., Greene, T. & Inker, L. A. Strengths and limitations of estimated and measured GFR. *Nat Rev Nephrol* **15**, 784, doi:10.1038/s41581-019-0213-9 (2019).
- 26 Heymsfield, S., Arteaga, C., McManus, C., Smith, J. & Moffitt, S. Measurement of muscle mass in humans: validity of the 24-hour urinary creatinine method *The American Journal of Clinical Nutrition* **37**, 478-494 (1983).
- 27 Crim, M. C., Calloway, D. H. & Margen, S. Creatine Metabolism in Men: Urinary Creatine and Creatinine Excretions with Creatine Feeding. *The Journal of Nutrition* **105**, 428-438 (1975).
- 28 Horber, F. F., Scheidegger, J. & Frey, F. J. Overestimation of Renal Function in Glucocorticosteroid Treated Patients. *European Journal of Clinical Pharmacology* **28**, 537-541 (1985).
- 29 Molitoris, B. A. & Reilly, E. S. Quantifying Glomerular Filtration Rates in Acute Kidney Injury: A Requirement for Translational Success. *Semin Nephrol* **36**, 31-41, doi:10.1016/j.semnephrol.2016.01.008 (2016).
- 30 Pavuluri, K. *et al.* Noninvasive monitoring of chronic kidney disease using pH and perfusion imaging. *Science Advances* **5**, doi:10.1126/sciadv.aaw8357 (2019).
- 31 Savage, N. Technology: Multiple Exposure. *Nature* **502**, 90-91 (2013).

REVIEWER COMMENTS

Reviewer #1 (Remarks to the Author):

The revised manuscript was very and thoroughly responsive to the criticisms. The novel features of the pH probe are now more clearly described and a model of kidney hydronephrosis is added. The work is much improved now.

With that said, Figure 7 including the text describing this is a bit confusing and should be improved, particularly panels d, f, h which are supposed to be cortex, medulla and pelvis but which highlight much of the same area of the right kidney, and also much of the same area in panels d and f for the left kidney. Why is this, shouldn't the areas be more differently highlighted? This is not so similar to what would be seen in Gad or CT studies of the uptake? Please explain a bit more including why there is a difference in left and right kidneys in panel h and why the regions look similar in spite of contrast in panels d, f. Is there a strong difference seen in the hydronephrosis for these two kidneys from T2 images for the rat in Fig 7 like seen in Fig S15a which looks more like a unilateral than a bilateral hydronephrosis? How different is the hydronephrosis from animal to animal from T2 as well, please mention.

Also, for Figure 6, why is there such a wide range of RBF and GFR for healthy rats as is seen in panels h and i, shouldn't this be similar for the rats in this group? Please discuss this a little as part of the comments on the figure. Is this large variation seen for other methods?

Reviewer #2 (Remarks to the Author):

I would like to thank the authors for significantly improving the paper.

I have a few comments, if the pelvic pH is the most significant indicator, I would imagine a simple urine test could be used to test the same thing. Have the authors tried measuring the urine pH? It's clearly a mix of both kidneys but it has been shown in sheep's that even O₂ can be indicative of renal disease in the bladder. A comment on this would be good.

Why do you need to do it simultaneously? Just to save time? Are all parameters equally important and thus could you provide the same diagnostic value with 1 or 2 of them. I am quite sure the authors

actually say that (pelvic pH etc.). So my question is actually still valid and is similar to the typical PET/MR discussion, what killer application requires the information to be acquired simultaneously (PET/MR we don't know still)? If the authors say, that it's just time saving and easier to acquire that is perfectly fine, but of course, if they have a clinical argument or even tested that the simultaneous acquisition has an improved specificity compared to the mentioned methods acquired individually, that would be a major finding and particularly important. A bit more discussion on this would be good.

Reviewer #3 (Remarks to the Author):

All comments have been adequately addressed.

Point-by-point response to reviewers' comments

We thank the reviewers for the thorough analysis of our revised manuscript and for their constructive suggestions to further improve our manuscript. We address all comments in a point-by-point manner in this response letter and we have modified our manuscript and supplement based upon the reviewers' suggestions. Changes in the manuscript with respect to the previous version are marked in yellow. Reviewer comments are written in italic letters and our respective responses are listed below with indentation. All references included in excerpts from the manuscript are indexed according to the manuscript, all others according to this letter.

Reviewer #1 (Remarks to the Author):

The revised manuscript was very and thoroughly responsive to the criticisms. The novel features of the pH probe are now more clearly described and a model of kidney hydronephrosis is added. The work is much improved now.

We thank the reviewer again for provoking thoughts which led to the overall improvement of our work and the way the results are presented to a broader audience.

With that said, Figure 7 including the text describing this is a bit confusing and should be improved, particularly panels d, f, h which are supposed to be cortex, medulla and pelvis but which highlight much of the same area of the right kidney, and also much of the same area in panels d and f for the left kidney. Why is this, shouldn't the areas be more differently highlighted?

First, we understand the reviewers point regarding the spatial cross-coverage of the pH compartments and like to describe both biological and technical reasons for this observation:

Biological reason for pH compartment map appearances in Fig. 7 d, f, h:

There is no sharp boundary between compartments. When looking at the anatomy of a kidney¹ (new Supporting Figure 1a), one can observe that the kidney is a complex-structured and well perfused organ which bears sophisticated branching networks of vessels to deliver large amounts of blood into the renal cortex (Supporting Figure 1a, left). As the cortex is therefore the best perfused area of the kidney, its pH is general very similar to the blood pH. This is reflected by the first pH compartment map (Fig. 7d), which mainly contains signal from blood vessels and the cortex. Since the entire kidney contains either renal blood vessels or cortex regions, the presence of this first compartment (Fig. 7d) across the entire kidney is well in line with anatomical structure. Taking a more detailed look at the kidney anatomy, its smallest functional units are the nephrons, which extend from the cortex through the medulla to the pelvis (Supporting Figure 1a, right). Being responsible for the filtration process, gradual acidification of the filtrate occurs during its passage through the nephron which allows regional assignment of the detected pH compartments to the nephron- and adjacent regions². Here, the glomeruli and the loop of Henle-surrounding vessels, with the latter extending throughout the entire medulla, also contain blood which explains why the first pH compartment (Fig. 7d) covers renal cortex and medulla regions (Supporting Figure 1a, right, blue-shaded area). Z-OMPD which enters the proximal tubules - and potentially resides longest in the loop of Henle - forms the second pH compartment (Supporting Figure 1a, right, green-shaded area), which shows slight acidification (=primary urine) due to the filtration-related acidification. As these tubules start in the cortex and with the loop of Henle reaching far down into the medulla, this explains why the second pH compartment also covers most of the kidney (Fig. 7f). The third and most acidic pH compartment is assignable to the collecting duct and the renal pelvis

(Supporting Figure 1a, right, red-shaded area), which also starts upper region of the medulla and extends down to the renal pelvis. As the collecting duct and the renal pelvis also cover significant areas of the kidney, this explains why also the third pH compartment shows considerable extension across the entire kidney (Fig. 7f). Nevertheless, to highlight, that there is some slight discrimination between the spatial coverage of the pH compartments, we overlaid the compartments from Fig. 7d, f, h (Supporting Figure 1b), which shows that the third compartment is slightly more confined to the center of the kidney, what matches the region assignments in a.

Supporting Figure 1 | Renal pH compartment localization. (a) Overview of the kidney anatomy (figure courtesy of the National Institute of Diabetes and Digestive and Kidney Diseases (NIDDK), National Institutes of Health¹; modified from copyright-free NIDDK-file) and a nephron in which assumed compartment localizations from pH imaging of Z-OMPD are indicated. (b) Overlap of all three pH compartments of Fig. 7d, f, h to show respective spatial extension. (c) Respective CSI slice positioning shown on a T_2 -weighted axial anatomical image where intention of multiple organ coverage results in some kidney areas being covered minorly while revealing cross-plane coverage of different kidney compartments within one voxel.

Technical reasons for pH compartment map appearances in Fig. 7 d, f, h:

The native spatial resolution ($4 \times 4 \times 4 \text{ mm}^3$) of the respective acquired CSI voxels (native voxel size indicated in Fig. 7d) relative to the size of the kidney structures as described above suggests that each voxel covers more than one pH-compartment per voxel (as seen in the spectra, Fig. 3c). Here, 4 mm native voxel size already extends beyond an entire renal pyramid, without considering a further decrease in the spatial resolution by the imaging point-spread function, thereby inevitably capturing all pH compartments as described in the previous section ("Biological reason"). Here, the point-spread-function of the CSI rather distorts representation of concave outlines of structures such as the medulla and the cortex (which appear C-shaped in coronal projection) in comparison to the convex pelvis (which appears ellipsoidal), leading to an overrepresentation of the first two in the central kidney region. Also, due to the kidney structure and anatomical compartment heterogeneity being approximately two orders smaller compared to our imaging resolution, we often cover the cortical/blood, medullary and the pelvic compartment in one voxel. This voxel is then lighting up in all three compartmental pH

maps. In addition, we want to point out again that our pH imaging technique so far relies on 2D CSI acquisitions, for which the coronal imaging slices of 4 mm thickness also often captures distinct kidney regions in perpendicular direction to the CSI slice (see also a following comment on kidney size appearance below and Supporting Figure 1c). Consequently, the limited spatial resolution of this CSI-acquisition makes it even more important to be able to distinguish the compartments on a sub-voxel level, as we can do by spectral assessment. As discussed in Fig. S7, the respective compartment maps are derived from peak fitting of the spectra, but there is no weighting on the intensity concluded in contrast to mean pH maps. We could imagine that a thinner slice or higher in-plane resolution might lead to a slightly stronger disentanglement of the pH compartments. There are potential ways to further increase the spatial resolution of pH maps, e.g. such as using faster acquisition techniques based on EPSI³, deuterating Z-OMPD to improve T₁ and achievable matrix size or, if the filtration/perfusion information is not required, solely imaging pH without prior perfusion imaging acquisition as performed for pH imaging in tumors (Fig. 4).

We also summed up these points by adding Supporting Figure 1 as Supplemental Figure S16 to the supplement (Supplement, P18-19, Fig. S16) and added a short paragraph to the results (P9, L50 – P10, L1):

All three compartments show large coverage of the kidney due to its anatomical complexity and limited scan resolution (Fig. S16).

and the discussion (P13, L20-23):

However, further improvement of spatial resolution and spatial coverage using a multi-slice or 3D acquisition technique might help to better spatially disentangle the pH compartments which have their origin in the microscopic heterogeneity and complexity of the kidney.

This is not so similar to what would be seen in Gad or CT studies of the uptake?

We would like to additionally comment on the difference of our imaging technique to Gd and CT studies as pointed out by the reviewer.

While for Gadolinium-based pH sensors, also pH imaging studies *in vivo* in healthy kidneys have been conducted already⁴, the respective kidney pH map (Fig. 6b in reference 4) doesn't allow to distinguish pH- nor anatomical compartments. Here, comparison to our perfusion and filtration imaging technique which dynamically captures Z-OMPD inflow and filtration in 3D is more appropriate, where Z-OMPD accumulation and filtration is qualitatively⁵ (Fig. 5d and Fig. 7a) and quantitatively^{6,7} (Fig. 6h, i) similar to e.g. Gd-DCE-MRI in rat kidneys.

However, it should be mentioned that, for hyperpolarized ¹³C studies, the achievable spatial resolution is generally lower compared to conventional Gd-based DCE-MRI or CT which potentially allows a higher spatial resolution for perfusion and filtration imaging of kidneys but at the cost of injecting potentially harmful Gd- or iodine-based contrast agents.

In conclusion, the "uptake" measure, meaning the perfusion and filtration assessment with Z-OMPD is similar to Gd/CT-based approaches while pH compartment maps from imaging with Z-OMPD provide a biological readout not accessible by Gd- or CT-based imaging techniques.

Please explain a bit more including why there is a difference in left and right kidneys in panel h and why the regions look similar in spite of contrast in panels d, f.

As the reviewer correctly observes, the third pH compartment area is smaller for the left kidney in Fig. 7h. The main reason for this is the CSI slice orientation and position:

Supporting Figure 2 | CSI slice positioning. In order to image pH in both the kidneys and the tumor-bearing adrenal glands, the CSI slice in MENX subjects did often not target the kidneys in their center. Three different axial images show the positioning of the CSI slice (green box) and the respective coverage of the kidneys (left + middle panel) as well as of the adrenal glands (right panel) is indicated by yellow arrows. For the left kidney, only a minor part of the pelvis is covered (left panel) while for the right kidney, the slice extends across several kidney regions (middle panel) while covering both adrenal glands (right panel).

The used FIDCSI sequence is a 2D slice acquisition with a thickness of 4 mm (Supporting Figure 2, green rectangles). In this study, the initial study goal was to simultaneously image the adrenal glands (adjacent to the kidneys) in addition to the kidneys, since the MENX rat model develops pheochromocytoma in these organs as already stated in the manuscript. As a result, positioning of the CSI slices was often a trade-off between coverage of all four organs (left kidney + right kidney: arrows in Supporting Figure 2, left and middle panel; left adrenal gland + right adrenal gland: arrows in right panel) and optimal coverage of the kidneys. This often led to more coverage of volume for one kidney compared to the other. More importantly, as indicated by the yellow arrows, for the left kidney (Supporting Figure 2, middle and right panel), a much smaller fraction of the renal pelvis region was covered compared to the medulla and cortex, leading to the minor appearance of the respective pH compartment in Fig. 7h. In addition, the following image (Supporting Figure 3) shows a different anatomical slice covering both kidneys optimally in this subject:

Supporting Figure 3 | Kidney morphology. T_2 -weighted coronal image slice of the kidneys of which pH compartment maps are shown in Fig. 7d, f, h. Both kidneys show comparable dilatation and severity of hydronephrosis.

One can see that the hydronephrosis is present in both kidneys with the pelvises being similarly strongly distorted, thereby excluding kidney- or pelvis size differences as a reason for the reduced compartment coverage.

We added a note regarding this aspect to the figure caption (P11, L1-2):

The left kidney and respective pH compartments in **d**, **f** and **h** exhibit a smaller cross-section due to inclined placement of the CSI slice.

Is there a strong difference seen in the hydronephrosis for these two kidneys from T2 images for the rat in Fig 7 like seen in Fig S15a which looks more like a unilateral than a bilateral hydronephrosis? How different is the hydronephrosis from animal to animal from T2 as well, please mention.

We thank the reviewer for making us aware that the slice representation of the hydronephrosis in Fig. S15a which shows one dilated kidney, and the respective dilated pelvis can cause confusion regarding the bilateral presence of this disease. Below (supporting figure 4) we show another slice from the same anatomical MRI acquisition on the same subject:

a

Supporting Figure 4 | Anatomical T_{2w} -image of hydronephrosis. Alternative coronal slice from the same RARE acquisition as the previously shown slice in Fig. S15a. Both kidneys show comparably strong dilatation of the renal pelvis, indicating hydronephrosis. Scale bar 10 mm.

In this slice, the hydronephrosis, manifested by the strongly dilated pelvis regions (white arrows) is clearly present in both kidneys to the same extent. We have also replaced the respective slice in Fig. S15a in the supplement by this image slice.

In addition, we have also generated an overview of anatomical T_2 -weighted images for two exemplary healthy subjects and the eight subjects (Supporting Figure 5, either one or two exemplary slices showing the kidneys are displayed) which were used to study hydronephrosis by imaging using hyperpolarized ^{13}C -labelled Z-OMPD for comparison:

Supporting Figure 5 | Overview of hydronephrosis in MENX subjects. Exemplary coronal T_2 -weighted images for all eight subjects (bottom) included in the study of hydronephrosis show bilateral dilatation of the renal pelvis (yellow arrows). Depending on the slices, additionally strongly dilated ureter can be observed as well. For comparison, respective images are also displayed for two healthy subjects (top) with regular kidney morphology (blue arrows).

In the healthy subjects, renal cortex, medulla are clearly visible as concentric rings around the renal pelvis which form an ellipsoid in the kidney center (blue arrows, healthy subjects 1+2). In sharp contrast, all studied subjects of hydronephrosis show non-uniform distorted and dilated renal pelvises (yellow arrows, hydronephrosis subjects 1-8) and dilated ureters (present in all subjects, visible here in subject 3, 7 and 8). While there is some variation in pelvis

dilatation between subjects, the severity appears to be uniform for both kidneys within all subjects. This is expected as the cause for the hydronephrosis, the increased calcium content in the blood and the high blood pressure which both together cause the underlying urinary tract obstruction leading to the hydronephrosis are systemic symptoms and therefore not specific to one kidney or the other. The frequency of the disease-causing pheochromocytomas is 100%.

This was also pointed out already in the manuscript and the presence of bilateral hydronephrosis is now further specified (P9, L40-42):

“These develop bilateral adrenal medullary tumors (frequency 100%), alias pheochromocytoma, which oversecrete catecholamines and lead to **bilateral** kidney hydronephrosis, thereby mimicking clinically observed disease aspects⁴⁵.”

Also, for Figure 6, why is there such a wide range of RBF and GFR for healthy rats as is seen in panels h and l, shouldn't this be similar for the rats in this group? Please discuss this a little as part of the comments on the figure. Is this large variation seen for other methods?

We agree with the reviewer, that both the RBF values (1 – 7 ml/min/ml_{tissue}) and the GFR values (0.5 – 3.5 ml/min) show a considerable variation. While this appears rather unexpected for a control group of healthy subjects at a first glance, comparison to measurements of these parameters in literature shows that this is well in line with previous studies. For example, the study on which we base our RBF quantification method and comparison of Z-OMPD to hyperpolarized ¹³C-urea⁷ also reports values ranging from 1.9 – 6.99 ml/min/ml_{tissue}, which is similar to our range. Also, tGFRs measured by DCE-MRI (1.2 – 5 ml/min) or Inulin (2.3 - 8 ml/min) agree well in range and absolute values with our study as already pointed out in the results (P9, L21-23 and P9, L26-27).

Apart from this study, large variations for measured GFRs have also been reported by other studies often relying on inulin which is considered a gold standard for GFR determination. Here, longitudinal studies in healthy rats showed variations of measured GFR up to a factor of 4^{8,9} while absolute values have been reported to range from 0.03 – 4.3 ml/min¹⁰. Also measurements using ¹²⁵I-iothalamate¹¹, ⁶⁸Ga-EDTA-PET¹² and sinistrin¹³ show variations of up to a factor of 10 for the determined GFR range.

While the rat age and the method uncertainty certainly contribute to these variations¹⁴ with the contrast agent even being a potential perturbing factor¹⁵, one cause could be a varying number of nephrons within a kidney¹⁶ for which single nephron glomerular filtration rates (SNGFR) were already determined and also show a variation of up to a factor of 2^{8,16}. Here, autoregulation of SNGFR occurs which is driven by blood pressure^{17,18} which itself is quite susceptible to stress and anaesthesia. Even large short-term variations of SNGFR due to simple saline-injections have been reported¹⁹.

Uncertainties in GFR measurements are not only observed in preclinical studies but also for clinical imaging-based methods using CT²⁰ where up to two standard deviations difference to blood tests are observed while both blood tests using ⁵¹Cr-EDTA²¹ or imaging with [¹⁸F]FDG-PET²² also show variations of up to a factor of two in healthy subjects.

Regarding the RBF measurements, we attribute some of our variation to partial volume effects due to the coarse spatial resolution (4 x 4 x 4 mm³) in comparison to the vessel size (descending aorta in a rat: 2 mm diameter). Another error source is bolus dispersion as we draw the input function from central vessels, not the renal arteries which are too small to be resolved and assume the sparse temporal sampling as another source of uncertainty²³. Nevertheless, also for this parameter variations of a factor 2 – 3 have been observed both for DCE-MRI and ASL-MRI in rats.^{6,24,25}

In summary, while we don't claim superior accuracy compared to existing method regarding RBF- and GFR-quantification, absolute values and ranges agree well with many studies in literature using different methods, thereby rendering perfusion imaging with Z-OMPD a valid quantification technique for renal blood flow and perfusion. We added some of these considerations in the discussion (P12, L33-43):

“However, strong variations between subjects for both GFR and RBF are observed. Here, one limitation is the low number of frames, resulting effectively in only 4–8 time points to capture flow and filtration dynamics. While this might result in some quantification error, large inter-subject variations for GFR (0.03 – 4.3 ml/min) have also been observed using the gold standard inuline⁷⁵⁻⁷⁷ while radiotracer^{78,79}, PET^{80,81} or sinistrin⁸² also show variation of up to a factor of 10 in healthy subjects. This might be explained by varying numbers of nephrons per kidney⁸³ which itself show single-nephron-GFR variations by up to a factor of 2^{75,83}. These can already occur on short time scales caused by blood-pressure-driven autoregulation^{84,85} or even simple saline injections⁸⁶. Variations similar to GFR have been reported for RBF⁸⁷⁻⁸⁹ while sparse temporal sampling, imaging resolution and bolus dispersion⁹⁰ might be the main source of uncertainty in our study. Nevertheless, absolute values and ranges overall agree well with literature, thereby validating our technique.”

Reviewer #2 (Remarks to the Author):

I would like to thank the authors for significantly improving the paper.

We thank the reviewer for raising important points, which we hope could all be addressed and helped us to substantially improve the manuscript.

I have a few comments, if the pelvic pH is the most significant indicator, I would imagine a simple urine test could be used to test the same thing. Have the authors tried measuring the urine pH? Its clearly a mix of both kidneys but it have been shown in sheep's that even O2 can be indicative of renal disease in the bladder. A comment on this would be good.

We agree with the reviewer that urine pH is another important parameter which can be easily accessed in a safe way in the clinic. We collected urine from healthy Wistar rats and MENX rats bearing hydronephrosis yielding the following result:

Supporting Figure 6 | Urine pH in subjects with hydronephrosis. Urine pH measured in healthy and MENX subjects does not indicate hydronephrosis, despite hydronephrotic urine tending towards slightly more alkaline pH values.

From our measurements of urine pH, there is no indication for renal disease, more specifically hydronephrosis while urine pH from this model shows a slight trend towards more alkaline values. While it appears intuitive that pelvic and urine pH should be closely related to each other, it has indeed been reported that hydronephrosis can manifest by an increased urine pH due to the obstruction and the inability to excrete acids²⁶. This obstruction in the in the ureter causes the pelvic pH to accumulate the backlog of acids which leads to its acidification while these acids are missing in the final urine, which consequently tends to more alkaline pH values. Thus, pelvic and urine pH are not directly indicative of each other in this disease type. Regarding the stronger variation for the urine pH is also known that urine pH is in general prone to various influences such as diet, fluid uptake or daytime^{27,28} and can therefore adapt values across a wide pH range. We added this new data to Fig. 7 as panel o and added the following sections to results and discussion:

Results (P11, L23-25): “Urine shows no relevant susceptibility to the strong acidification in the renal pelvis (p = 0.55) despite a slight trend for more alkaline pH values due to obstruction-related inability of acid excretion⁶⁷.”

Discussion (P13, L6-7): “While these parameters prove to be more sensitive to the present disease features compared to standard blood- or urine testing and routine MRI methods, ...”

Why do you need to do it simultaneously? Just to save time? Is all parameters equally important and thus could you provide the same diagnostic value with 1 or 2 of them. I am quite sure the authors actually say that (pelvic pH etc.). So my question is actually still valid and is similar to the typical PET/MR discussion, what killer application requires the information to be acquired simultaneously (PET/MR we don't know still)? If the authors say, that its just time saving and easier to acquire that is perfectly fine, but of course, if they have a clinical argument or even tested that the simultaneous acquisition has an improved specificity compared to the mentioned methods acquired individually, that would be a major finding and particularly important. A bit more discussion on this would be good.

We certainly agree with the reviewer that for application in clinical routine, saving time is one of the strongest needs. In fact, as the reviewer brings up the question about the competition between PET/MR and PET/CT, the reduced acquisition time of PET/CT and ease of use compared to PET/MR while providing comparable diagnostic information is one of the major reasons why it is more routinely used in the clinic.

From the perspective of decision making and the optimization of diagnostic value, if several parameters are measured simultaneously, clinicians don't have to decide which parameter is the most important and thus gets assessed first. Instead, for each acquisition, clinicians get a comprehensive dataset characterizing renal function not only based on one, but several (here up to five) parameters. So even though potentially just one parameter indicates disease, the knowledge of four other parameters being physiological also provides diagnostic value.

Regarding a potential "killer application" (we prefer the term "hero experiment") for simultaneous acquisition of these parameters, we envision the combined set of parameters to be a specific "fingerprint" of the kidney function and disease type. For hydronephrosis, we showed pathological alterations in GFR and pH_{Pelvis} values, where we also agree that in this case the pelvic pH is of most use, while the residual parameters (RBF, pH_{Cortex}, pH_{Medulla}) remain regular. Nevertheless, our prospective idea is that different kidney diseases exhibit their own "fingerprints", rendering this protocol a "one size fits all approach" for kidney function evaluation and disease detection. To validate this approach, similar studies as the one presented in this work on hydronephrosis should be conducted in different disease models of the kidney to identify those disease fingerprints and investigate whether these parameters are enough for disease differentiation as already pointed out in the discussion (P13, L8-10). Defining a clinical "killer application" just based on one study so far is in our opinion too ambitious.

Combining the parameters in the described "fingerprint" however is difficult if the parameters are not being measured simultaneously, since different methods and tests introduce their own biases which adds up to the biases stemming from different timing (physiological changes of patient: fatigue, nutritional status, circadian rhythm, disease progression, etc.). Therefore, we believe that the simultaneous acquisition is crucial for reproducibility and reliability of the results. In addition, clinical translation of Z-OMPD is facilitated by the internal frequency referencing and the dual-use of the C₁-label for perfusion and pH imaging. As a result, no co-injection of additional agents such as urea is needed which would otherwise require additional toxicological studies and complicate the probe preparation protocol.

In addition to the clinical potential of the simultaneous acquisition, its value was already claimed in preclinical studies²⁹ where superior accuracy was observed compared to studies with delays between measurements of different parameters³⁰:

“One possible reason the DCE-MRI and FITC-sinistrin measurements were in better agreement in this study is that they were acquired simultaneously, whereas in the previous study by Sadick et al. the two measurements were acquired on consecutive days.”

While the “fingerprint” aspect (P13, L6-8) and an envisioned “killer application” (P13, L31-34) are already pointed out in the discussion, we also added a short paragraph on the advantage of simultaneous acquisition methods (P13, L13-16):

“Here, in addition to the short scan time being a major advantage, a comprehensive set of kidney parameters is provided simultaneously, which omits the need for selection of one parameter for function diagnosis. Importantly, simultaneous assessment of multiple parameters is considered to provide more robustness and reliability^{92,93} as serial tests might corrupt each other⁹⁴.”

References:

- 1 National Institute of Diabetes and Digestive and Kidney Diseases, N. I. o. H.
- 2 Gaohua, L., Miao, X. & Dou, L. Crosstalk of physiological pH and chemical pKa under the umbrella of physiologically based pharmacokinetic modeling of drug absorption, distribution, metabolism, excretion, and toxicity. *Expert Opin Drug Metab Toxicol* **17**, 1103-1124, doi:10.1080/17425255.2021.1951223 (2021).
- 3 Topping, G. J. et al. Acquisition strategies for spatially resolved magnetic resonance detection of hyperpolarized nuclei. *MAGMA* **33**, 221-256, doi:10.1007/s10334-019-00807-6 (2020).
- 4 Raghunand, N., Howison, C., Sherry, A. D., Zhang, S. & Gillies, R. J. Renal and systemic pH imaging by contrast-enhanced MRI. *Magn Reson Med* **49**, 249-257, doi:10.1002/mrm.10347 (2003).
- 5 Pedersen, M. et al. in *Preclinical MRI of the Kidney: Methods and Protocols* (2021).
- 6 Zimmer, F. et al. Quantitative renal perfusion measurements in a rat model of acute kidney injury at 3T: testing inter- and intramethodical significance of ASL and DCE-MRI. *PLoS One* **8**, e53849, doi:10.1371/journal.pone.0053849 (2013).
- 7 Ostergaard Mariager, C. et al. Can Hyperpolarized ¹³C-Urea be Used to Assess Glomerular Filtration Rate? A Retrospective Study. *Tomography* **3**, 146-152, doi:10.18383/j.tom.2017.00010 (2017).
- 8 Andreucci, V. E., Dal Canton, A. & Corradi, A. Reliability of Radioactive Inulin as a Marker of Glomerular Filtration Rate in the Rat. *Pflügers Archive* **350**, 347-358, doi:10.1007/BF00592643 (1974).
- 9 Laurent, D., Poirier, K., Wasvary, J. & Rudin, M. Effect of essential hypertension on kidney function as measured in rat by dynamic MRI. *Magn Reson Med* **47**, 127-134, doi:10.1002/mrm.10034 (2002).
- 10 Besseling, P. J. et al. A plasma creatinine- and urea-based equation to estimate glomerular filtration rate in rats. *Am J Physiol Renal Physiol* **320**, F518-F524, doi:10.1152/ajprenal.00656.2020 (2021).
- 11 de Vries, P. A. M., Navis, G., de Boer, E., de Jong, P. E. & de Zeeuw, D. A method for accurate measurement of GFR in conscious, spontaneously voiding rats. *Kidney Int* **52**, 244-247, doi:10.1038/ki.1997.327 (1997).
- 12 Ding, Y. et al. Glomerular filtration rate calculation based on ⁶⁸Ga-EDTA dynamic renal PET. *American Journal of Nuclear Medicine and Molecular Imaging* **12**, 54-62 (2022).
- 13 Schock-Kusch, D. et al. Transcutaneous measurement of glomerular filtration rate using FITC-sinistrin in rats. *Nephrol Dial Transplant* **24**, 2997-3001, doi:10.1093/ndt/gfp225 (2009).

- 14 Fleck, C. Determination of the Glomerular Filtration Rate (GFR): Methodological Problems, Age-Dependence, Consequences of Various Surgical Interventions, and the Influence of Different Drugs and Toxic Substances. *Physiological Research* **48**, 267-279 (1999).
- 15 Ueda, J., Nygren, A., Hansell, P. & Erikson, U. Influence of Contrast Media on Single Nephron Glomerular Filtration Rate in Rat Kidney. *Acta Radiologica* **33**, 596-599, doi:10.1080/02841859209173221 (1992).
- 16 Bech, S. K. *et al.* The number of glomeruli and pyruvate metabolism is not strongly coupled in the healthy rat kidney. *Magn Reson Med* **87**, 896-903, doi:10.1002/mrm.29025 (2022).
- 17 Häberle, D. A. *et al.* Autoregulation of the glomerular filtration rate and the single-nephron glomerular filtration rate despite inhibition of tubuloglomerular feedback in rats chronically volume-expanded by deoxycorticosterone acetate. *Pflügers Archiv* **416**, 548-553 (1990).
- 18 Navar, L. G., Burke, T. J., Robinson, R. R. & Clapp, J. R. Distal tubular feedback in the autoregulation of single nephron glomerular filtration rate. *The Journal of Clinical Investigation* **53**, 516-525, doi:10.1172/JCI107585 (1974).
- 19 Gertz, K. H., Brandis, M., Braun-Schubert, G. & Boylan, J. W. The effect of saline infusion and hemorrhage on glomerular filtration pressure and single nephron filtration rate. *Pflügers Archive* **310**, 193-205 (1969).
- 20 Hackstein, N., Wiegand, C., Rau, W. S. & Langheinrich, A. C. Glomerular filtration rate measured by using triphasic helical CT with a two-point Patlak plot technique. *Radiology* **230**, 221-226, doi:10.1148/radiol.2301021266 (2004).
- 21 Kang, Y. K. *et al.* Quantitative Single-Photon Emission Computed Tomography/Computed Tomography for Glomerular Filtration Rate Measurement. *Nucl Med Mol Imaging* **51**, 338-346, doi:10.1007/s13139-017-0491-8 (2017).
- 22 Geist, B. K. *et al.* Assessing the kidney function parameters glomerular filtration rate and effective renal plasma flow with dynamic FDG-PET/MRI in healthy subjects. *EJNMMI Res* **8**, 37, doi:10.1186/s13550-018-0389-1 (2018).
- 23 Calamante, F. Arterial input function in perfusion MRI: a comprehensive review. *Prog Nucl Magn Reson Spectrosc* **74**, 1-32, doi:10.1016/j.pnmrs.2013.04.002 (2013).
- 24 Jiang, K., Tang, H., Mishra, P. K., Macura, S. I. & Lerman, L. O. Measurement of Murine Single-Kidney Glomerular Filtration Rate Using Dynamic Contrast-Enhanced MRI. *Magn Reson Med* **79**, 2935-2943, doi:10.1002/mrm.26955 (2018).
- 25 Zollner, F. G., Zimmer, F., Klotz, S., Hoeger, S. & Schad, L. R. Renal perfusion in acute kidney injury with DCE-MRI: deconvolution analysis versus two-compartment filtration model. *Magn Reson Imaging* **32**, 781-785, doi:10.1016/j.mri.2014.02.014 (2014).
- 26 Chandar, J., Abitbol, C., Novak, M., Zilleruelo, G. & Strauss, J. Abnormal urinary acidification in infants with hydronephrosis. *Pediatric Nephrology* **13**, 315-318 (1999).
- 27 Tannehill-Gregg, S. H. *et al.* Strain-related differences in urine composition of male rats of potential relevance to urolithiasis. *Toxicol Pathol* **37**, 293-305, doi:10.1177/0192623309332990 (2009).
- 28 Boswald, L. F., Matzek, D., Kienzle, E. & Popper, B. Influence of Strain and Diet on Urinary pH in Laboratory Mice. *Animals (Basel)* **11**, doi:10.3390/ani11030702 (2021).
- 29 Zollner, F. G. *et al.* Simultaneous measurement of kidney function by dynamic contrast enhanced MRI and FITC-sinistrin clearance in rats at 3 tesla: initial results. *PLoS One* **8**, e79992, doi:10.1371/journal.pone.0079992 (2013).
- 30 Sadick, M. *et al.* Two non-invasive GFR-estimation methods in rat models of polycystic kidney disease: 3.0 Tesla dynamic contrast-enhanced MRI and optical imaging. *Nephrol Dial Transplant* **26**, 3101-3108, doi:10.1093/ndt/gfr148 (2011).

REVIEWERS' COMMENTS

Reviewer #1 (Remarks to the Author):

ACcept, satisfied with response

Reviewer #2 (Remarks to the Author):

Only one last change needed. Please change the multi papillary kidney to a uni papillary kidney illustration in the new figure.

Point-by-point response to reviewers' comments

We thank the reviewers for the thorough analysis of our revised manuscript and for principally agreeing with its current form. We address the remaining comment in this response letter and we have modified our manuscript and supplement based upon the reviewers' suggestions. Changes in the manuscript with respect to the previous version are marked in yellow. Reviewer comments are written in italic letters and our respective responses are listed below with indentation.

Reviewer #2 (Remarks to the Author):

Only one last change needed. Please change the multi papillary kidney to a uni papillary kidney illustration in the new figure.

We thank the reviewer for the observation that we displayed a multilobar kidney in explanatory figures. We acknowledge that rodents have a unilobar kidney and changed supplementary figure S16a as well as manuscript figure 5a accordingly.